# Joint inference of CFC lifetimes and banks suggests previously unidentified emissions

Megan Lickley [1✉], Sarah Fletcher [2], Matt Rigby [3] & Susan Solomon [1]

Chlorofluorocarbons (CFCs) are harmful ozone depleting substances and greenhouse gases. CFC production was phased-out under the Montreal Protocol, however recent studies suggest new and unexpected emissions of CFC-11. Quantifying CFC emissions requires accurate estimates of both atmospheric lifetimes and ongoing emissions from old equipment (i.e. 'banks'). In a Bayesian framework we simultaneously infer lifetimes, banks and emissions of CFC-11, 12 and 113 using available constraints. We find lifetimes of all three gases are likely shorter than currently recommended values, suggesting that best estimates of inferred emissions are larger than recent evaluations. Our analysis indicates that bank emissions are decreasing faster than total emissions, and we estimate new, unexpected emissions during 2014-2016 were 23.2, 18.3, and 7.8 Gg/yr for CFC-11, 12 and 113, respectively. While recent studies have focused on unexpected CFC-11 emissions, our results call for further investigation of potential sources of emissions of CFC-12 and CFC-113, along with CFC-11.

[1] Department of Earth, Atmospheric, and Planetary Sciences, Massachusetts Institute of Technology, Cambridge, MA, USA. [2] Civil and Environmental Engineering, Stanford University, Stanford, CA, USA. [3] School of Chemistry, University of Bristol, Bristol, UK. ✉email: mlickley@mit.edu

The Montreal Protocol entered into force in 1989 and led to the phase-out of industrial production of CFCs by 2010. This global action reduced emissions of Ozone Depleting Substances (ODSs) and avoided massive worldwide ozone losses[1]. There are now signs that the Antarctic ozone hole is beginning to heal[2,3]. While the phase-out of ODSs has led to decreases in their atmospheric abundance, recent studies have pointed to an observed and unexpected delay in the decrease of CFC-11 concentrations, indicating ongoing emissions of this gas that may be linked to illicit production[4,5]. CFC-11 is the most abundant source of chlorine in the atmosphere and a potent greenhouse gas. Limiting its production and emissions is, therefore, critical for both ozone recovery and to safeguard the planet against climate change. Quantifying illicit emissions of CFCs has been limited by uncertainties in their underlying emissions from old equipment (i.e., banks) with uncertain leak rates (release fractions), as well as the lifetimes of these molecules in the atmosphere. Joint consideration of atmospheric lifetimes together with observed changes in atmospheric concentrations hence provide our best estimate of global total emissions (i.e., both bank and illicit emissions). Earlier work has focused on quantifying the size of CFC banks and potential contributions to future emissions[6]; here we focus on jointly estimating lifetimes of CFC-11, 12, and 113 as well as the magnitude and sources of global emissions of these gases using all available information in a probabilistic framework.

While there are several lifetime estimation methods, (e.g., tracer–tracer or observation-based methods, see SPARC[7] for details), two of the most widely used techniques are discussed here. The first makes use of 3D chemistry climate models (CCM) that integrate chemical sink loss frequencies and calculated global distributions[8]. Results from six 3D CCMs and one 2D CCM were made available through the Stratosphere-troposphere Processes and their Role in the Climate (SPARC) modeling effort[7,9], a project of the World Climate Research Programme. CCMs use observed surface concentrations as input, and the accuracy of their modeled lifetimes largely depends on their ability to accurately model atmospheric transport and chemical loss processes. It takes ~3–7 years for air to be transported from the source of emission to the upper atmospheric loss region and back. SPARC models thus predict a decreasing trend in ODS lifetimes, reflecting the disequilibrium between tropospheric and stratospheric concentrations when tropospheric concentrations are increasing. These forward CCM simulations thus estimate "time-varying" lifetimes based on observed variations of ODS concentrations. A near "steady-state" lifetime is achieved a few years after tropospheric concentrations stop their rapid rise. The second approach referred to as the "inverse modeling method", infers lifetimes in a Bayesian framework using near-surface mole fraction measurements and, typically, fixed estimates of emissions as inputs to an atmospheric model[10]. These different approaches include different assumptions and over the years have led to wide ranges in estimated steady-state lifetimes (see Table 1 of present manuscript and Table 6.1 of SPARC[7]), and therefore large differences in inferred emissions.

While knowledge of atmospheric lifetimes allows for emissions to be inferred from changes in observed concentrations, further analysis is required to partition the source of those emissions from new production versus emissions from banks. The former would represent a breach of the Montreal Protocol while the latter remains permitted at this time, highlighting the policy importance of quantifying bank emissions. There are various approaches to quantifying banks and their emissions. A top-down approach makes use of observationally-derived emissions along with assumptions regarding production and lifetimes[11]. The sizes of banks are then estimated as the cumulative difference in production and emissions over time.

Some emissions, such as leakage during production, occur quickly (here referred to as direct emissions), while some are delayed by storage in banks. Figure 1 illustrates how different plausible lifetime and production assumptions can propagate into large differences in top-down inferred banks. With respect to lifetimes, we compare an assumed time-varying lifetime based on mean SPARC modeled values with an assumed steady-state lifetime of 52 years, as cited in WMO (2018), Table A-1[3], to infer historical emissions. Hence, Fig. 1 underscores the importance of using appropriate time-varying lifetimes in emissions and bank calculations. Figure 1 further illustrates how small differences in assumed bank release fractions (i.e., leakage rate from banks) can propagate into uncertainties in the source of emissions (i.e., whether emissions come from banks versus direct emissions from new production). Figure 1 serves to illustrate how different seemingly reasonable assumptions about lifetimes, production, and release fractions can both match observed concentrations and lead to very different conclusions about the quantity and sources of emissions, with some combinations of assumptions being unphysical (i.e., producing negative direct emissions). Further examples are presented in the SI.

An alternative bottom-up approach to estimating banks and their emissions makes use of industry reported production data along with a careful tallying of multiple equipment types and estimated release fractions over time[12,13]. While this approach is less reliant on indirect inference of bank magnitudes than the top-down approach, it requires accurate and complete reporting and does not make full use of all available data, such as atmospheric observations.

In recent work, Lickley and colleagues[6] take a Bayesian Parameter Estimation (BPE) approach, which makes use of observationally-derived emissions to apply inference to a deterministic bottom-up bank simulation model on a gas-by-gas basis for CFC-11, 12, and 113. This bank estimation approach incorporates the widest range of constraints to date, including observations, reported production data, and previously published release rates partitioned by equipment types and their respective

**Table 1 Posterior atmospheric lifetime estimates for the unexpected emissions scenario and a comparison with previously published estimates.**

|  | BPE-derived median posterior lifetimes (p2.5, p97.5) | | WMO (2003)/ IPCC (2001) | SPARC recommended steady state (most likely 2-sigma CI) | Rigby et al. (2013) (1-sigma CI) |
|---|---|---|---|---|---|
|  | 2010 lifetimes | Time-averaged lifetimes |  |  |  |
| CFC-11 | 49.1 (44.1, 54.6) | 54.0 (47.5, 62.7) | 45 | 52 (43, 67) | 52 (45, 61) |
| CFC-12 | 84.7 (76.7, 95.5) | 93.2 (81.7, 111.5) | 100 | 102 (88, 122) | 112 (95, 136) |
| CFC-113 | 79.9 (74.0, 88.7) | 89.3 (79.6, 104.7) | 85 | 93 (82, 109) | 109 (97, 124) |

Results shown from the present study include a 95% CI (p2.5, p97.5).

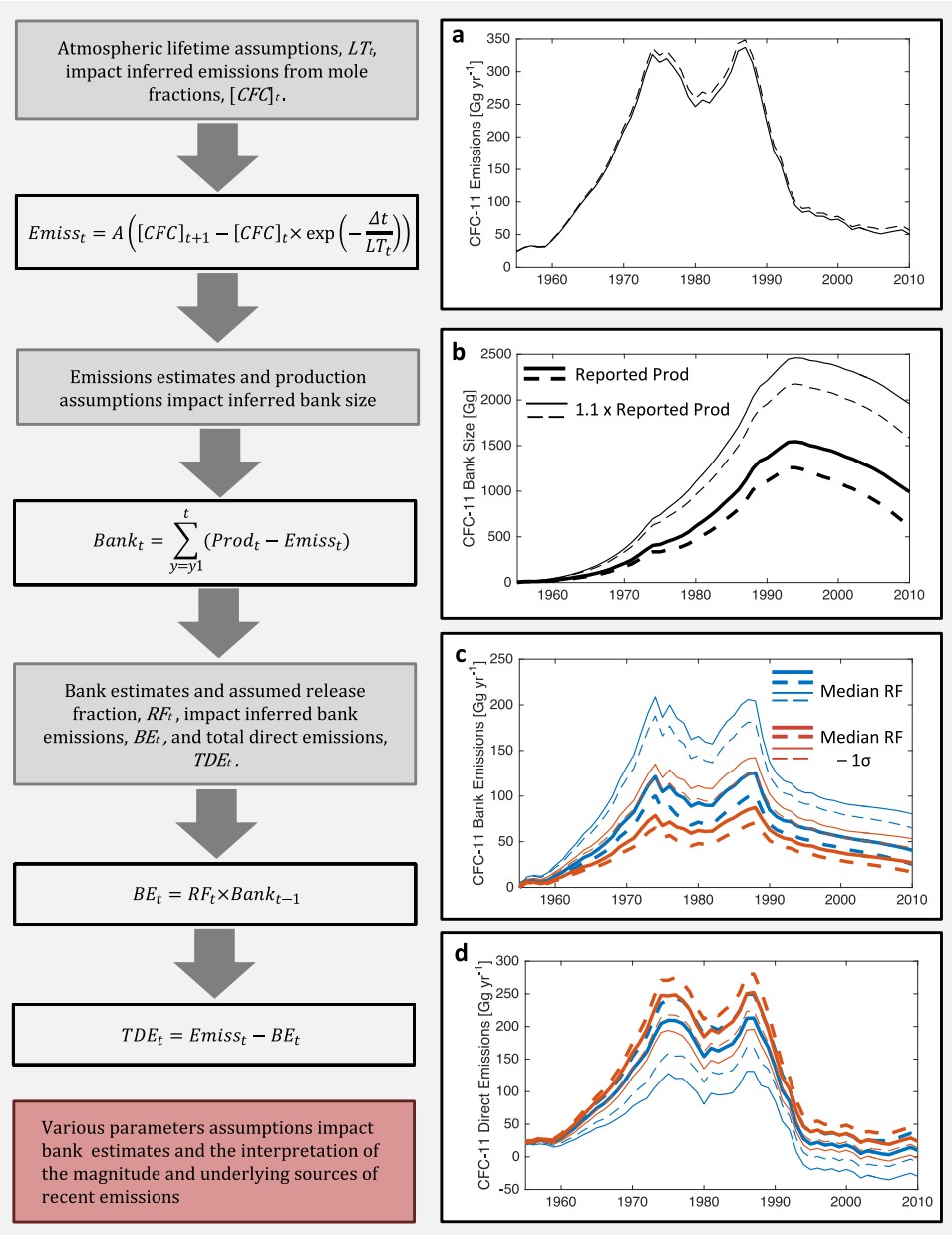

**Fig. 1 Impact of parameter assumptions on banks and emissions estimates.** Panel (**a**) illustrates the impacts of two atmospheric lifetime (LT) assumptions on inferred total emissions (Emiss) from observed concentrations, while Panels (**b**–**d**) provide illustrative examples to show the sensitivities of the breakdown of sources of those emissions to components from banks (BE) versus direct emissions (TDE). The solid lines throughout correspond to quantities derived assuming the SPARC multi-model mean time-varying lifetime (with a time-averaged lifetime of 62 years), and the dashed lines correspond to an assumed constant lifetime of 52 years, adopted as the steady state lifetime in the work of Rigby and colleagues[5] and WMO (2018)[3]. Panel (**a**) shows the emissions estimate using the first equation. The parameter, $A$, is a constant that converts units of atmospheric concentrations to units of emissions. Panel (**b**) shows the bank estimates for the two lifetime scenarios and two illustrative production scenarios; bank estimates using reported production are shown using the thicker lines, and the thinner lines correspond to a production scenario that is 10% larger than reported. Panel (**c**) shows inferred bank emissions for the lifetime scenarios and production scenarios described above. The colors of the lines indicate different bank release fraction (RF) estimates. Blue lines correspond to bank emissions for the median of the prior release fractions distributions used in the analysis from this paper. The red lines correspond to the median $-1\sigma$ of the prior release fraction distribution. Panel (**d**) shows the broad range of resulting inferred direct emissions, with the lines corresponding to the same scenarios used in Panel (**c**).

uncertainties. That work found banks of CFC-11 and 12 were likely larger than previous scientific assessments suggested, although CFC-11 banks alone could not account for the recent uptick in global emissions that Montzka and colleagues[4] inferred from observed concentrations.

Here, we extend the BPE model from Lickley et al. (2020)[6] to jointly rather than individually infer lifetimes and their

uncertainties for CFC-11, 12, and 113 and address correlations across molecules. In doing so, we account for the common atmospheric transport and chemical loss processes that govern the lifetimes of CFC-11, 12, and 113. Evidence of correlated lifetimes comes, for example, from the SPARC set of CCMs, where a model producing lower than average CFC-11 lifetimes also produces lower than average CFC-12 and 113 lifetimes. We

find lifetimes of CFC-12 and 113 are significantly shorter than current SPARC recommended values, implying that observationally derived emissions are likely larger than current best estimates. We find posterior estimates of non-bank emissions during 2014–2016 were 23.2 for CFC-11, 18.3, and 7.8 Gg/yr for CFC-11, 12, and 113, respectively, which are substantially larger than previously identified sources. This calls for further investigation of potential sources of unexpected emissions not only for CFC-11, but also of CFC-12 and CFC-113.

## Results

**Modeling framework**. To derive lifetimes, emissions and their respective sources, we build on the Bayesian Parameter Estimation approach to modeling banks and emissions developed by Lickley and colleagues[6]. The basic framework develops a bottom-up simulation model that simultaneously models banks, emissions (partitioned into bank emissions versus direct emissions), and mole fractions over time. The simulation model relies on inputs including reported production of each equipment type, estimates of release fractions and their respective uncertainties, and lifetime estimates developed using the SPARC CCM time-varying values. We develop prior distributions for each of the input parameters using previously published values. We use SPARC CCM modeled time-varying lifetimes to construct joint lifetime priors for CFC-11, 12, and 113 that reflect the lifetime uncertainties and correlations both across time and gases in the BPE model. Sampling from these prior distributions, we also develop priors for each of the outputs of the simulation model (i.e., banks, bank emissions, direct emissions, and mole fractions). Observed mole fractions are treated as observations in Bayes' Theorem and used to estimate the posteriors of each of the inputs and outputs of the simulation model (See "Methods" for more details). Adding these new constraints to the BPE model allows us to simultaneously infer lifetimes along with total emissions, bank size, and bank emissions over time. In this way, we can better quantify and partition recent emissions into bank emissions versus unexpected sources using all available information.

**Mole fraction estimates**. Figure 2 shows the observations, prior and posterior distributions of mole fraction estimates for CFC-11, 12, and 113. The observations are within the range of inferred posterior model predictions of the data for all gases throughout the whole time period (1955–2010). Posterior residuals (i.e., $D_j - M_j$ from Eq. (8); see "Methods") are shown in the supplement (Fig. S1) and appear normally distributed around zero.

**Emissions estimates and comparison**. Figure 3 shows prior and posterior emissions estimates, calculated using the parameter-based estimate of emissions (i.e., Equation (2); see "Methods"), compared to an observationally-derived estimate, where emissions are inferred using mole fractions and atmospheric lifetimes (i.e., solving for $E_{j,t}$ using Eq. (1) of "Methods"). This provides a useful contrast to previous emissions comparisons from Lickley et al. (2020)[6], which adopted both a time-varying lifetime equal to the SPARC multi-model mean, and a fixed lifetime scenario equal to the time-average mean of SPARC multi-model mean. Compared to Lickley et al. (2020), the posterior emissions in Fig. 2 show a closer match between parameter-based and observationally derived emissions as well as lower uncertainties, which can be explained in part by lifetimes being inferred instead of assumed. By jointly inferring lifetimes, observationally-derived emissions are represented as a distribution instead of fixed values, and our simulated emissions and mole fractions are assumed to have lower uncertainty than in Lickley et al. (2020), where lifetimes were fixed. This leads to a smaller posterior distribution in

parameter-based emissions estimates that more closely matches the observationally-derived estimates. We note that WMO 2018 emissions estimates for CFC-11 are within the uncertainty of our posterior observationally-derived emissions. For CFC-12 and 113, however, WMO 2018 emissions fall below the range of our posterior observationally-derived emissions towards the end of the simulation period, a result of posterior lifetimes being shorter for these two gases than what is assumed in WMO 2018[3].

**Lifetime estimates and comparison**. Figure 4 shows the time-varying prior and posterior lifetime estimates for CFC-11, 12, and 113. The SPARC multi-model mean is included for comparison. This figure shows that the median joint posterior lifetime estimates are shorter than the SPARC multi-model mean values throughout the time period, though within the 2-sigma range, for all three species. Note that it is not surprising that similar differences with the posterior are found for all species, due to the strong correlation assumed between model lifetimes. If the BPE is instead run without assuming lifetime dependency across molecules, then the result looks substantially different (Supplementary Fig. 2). In this case, CFC-12 posterior lifetimes are significantly lower than the SPARC multi-model mean, whereas CFC-11 and CFC-113 posterior estimates are more similar to their priors than the joint posterior estimates. This finding shows that our results are strongly dependent upon the inclusion of CCM lifetime uncertainties that are highly correlated between species. This correlation is obtained from the SPARC model ensemble, reflecting that the lifetimes of CFC-11, 12, and 113 are governed by similar physical and chemical processes as noted above. Thus, we argue that its inclusion makes the best use of the available evidence. The uncorrelated version of the BPE model (i.e., not including the dependencies across molecules in the lifetime priors) leads to a higher estimated lifetime and lower estimated Direct Emissions for CFC-11, and thus higher banks. Considering the joint dependency across molecules, inferring lifetimes instead of assuming them, and therefore reducing the assumed variance in the likelihood function accounts for much of the difference between bank estimates found here and Lickley et al. (2020) (See Supplementary Figs. 3 and 4).

Our BPE analysis provides an estimate of time-varying lifetimes and emissions up until 2010, which is the end of the time period for the SPARC CCM simulations. For emissions beyond 2010, we adopt a constant lifetime using posterior emissions values from 2010 (see "Methods"). The posterior time-averaged lifetime and 2010 lifetime estimates are shown in Table 1 along with comparisons with previously published estimates, importantly the SPARC recommended steady-state lifetime. To derive their recommended lifetimes, SPARC takes a weighted average of various estimation methods where the weights reflect the level of uncertainty in each method (see SPARC Chapter 6[7] for details). Inverse-modeling and CCM derived lifetimes are among the methods used for all of CFC-11, 12, and 113. For CFC-11 and 12, satellite observation-derived lifetimes are also included. The tracer–tracer method provides values for CFC-12 and 113 based on an assumed CFC-11 lifetime. While our CFC-11 2010 lifetime estimates agree with the SPARC recommended lifetime estimates, our CFC-12 and 113 estimates are outside the 2-sigma range that SPARC estimated to be most likely (shown in Table 1), but within their possible 2-sigma range ((78, 151) for CFC-12 and (69, 138) for CFC-113; see Chapter 6 in SPARC[7]). Why would our CFC-11 lifetime estimate agree with SPARC recommended lifetimes, but not CFC-12 and 113? We attribute this to two factors. The first is that the SPARC recommended values do not explicitly account for the inter molecule correlations exhibited by the CCM modeled lifetimes.

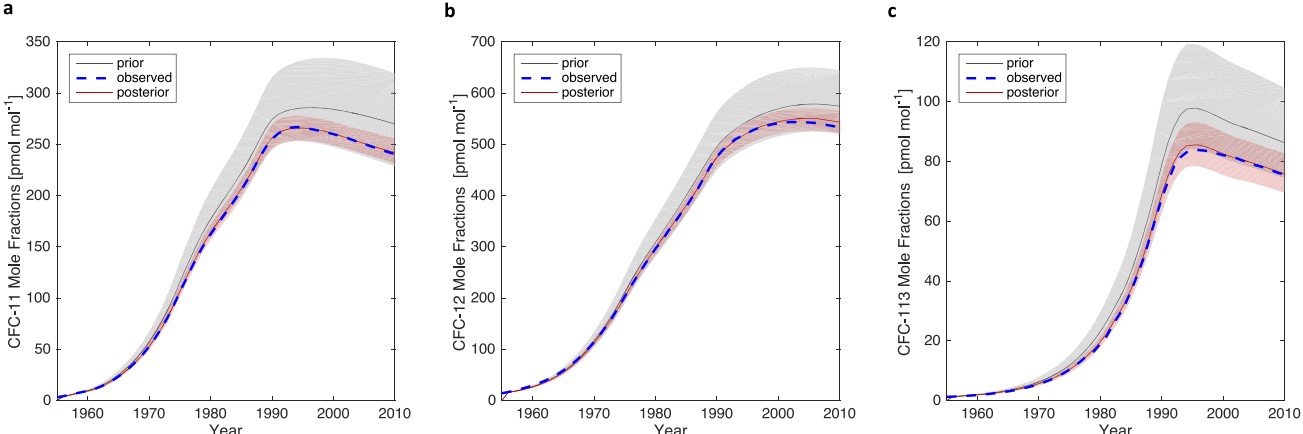

**Fig. 2 Mole fraction estimates.** The grey line and shaded region indicate median and 95% CI of the prior distribution for (**a**) CFC-11 (**b**) CFC-12, and (**c**) CFC-113. The dashed blue lines are observed concentrations and the red line and shaded region indicate the median and 95% CI for the posterior distributions.

**Fig. 3 Prior and posterior emissions and comparisons.** Emissions comparisons of observationally-derived (red) and parameter-based (grey) inferred emissions for the priors (**a**–**c**) and posteriors (**d**–**f**). The solid lines indicate the median and the shaded region indicates the 95% CI for both prior and posterior distributions. The uncertainties in observationally-derived emissions reflect the role of uncertainties in lifetimes and thus are updated in the posterior to reflect the lifetime posterior estimates. The dashed blue line indicates global emissions estimates from WMO (2018)[3], provided for comparison.

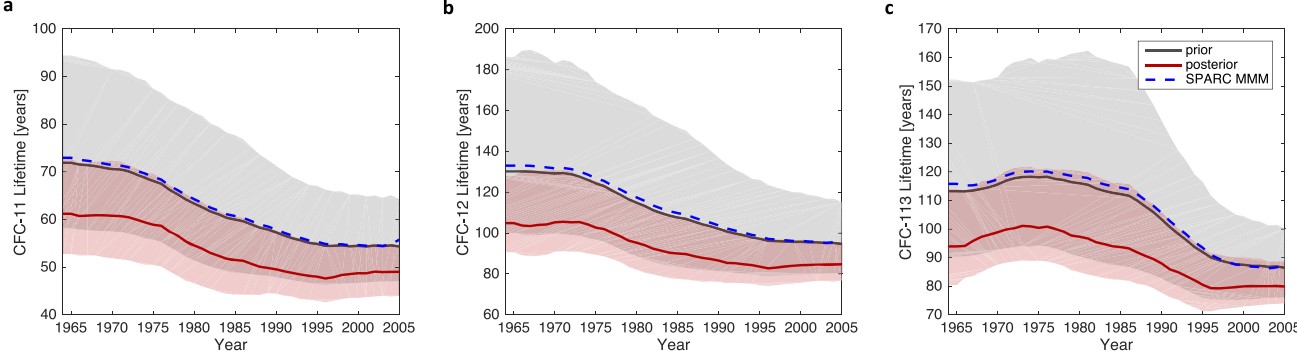

**Fig. 4 Time varying lifetime distributions.** The black line and grey shaded region indicate the median and 95% CI of the prior lifetimes, derived from SPARC modeled values for (**a**) CFC-11 (**b**) CFC-12, and (**c**) CFC-113. In each panel, the dashed blue line indicates the SPARC multi-model mean. The red line and shaded region indicate the median and 95% CI of the posterior lifetime distributions.

While the CCM derived lifetimes are a component of the weighted average, it is only the multi-model mean that is included and the inter-molecule correlations exhibited by the CCMs are not explicitly modeled in the weighted average. The second (which is a key limitation in earlier studies that use inverse-modeling to derive lifetimes, including SPARC) is that the recommended lifetimes depend on emissions assumptions, whereas our analysis jointly infers emissions along with lifetimes. We expect that the SPARC prescribed emissions for CFC-12 and 113 are biased low relative to CFC-11 for the inverse-modeling component of their estimate, contributing to this difference in recommended values. This is supported by the relatively long lifetimes for CFC-12 and 113 coming from the inverse-modeling component of the weighted average (see Table 6.1 from Chapter 6, SPARC[7]).

**Emissions estimates from banks versus new production**. We adopt the 2010 posterior lifetime distributions from our analysis, along with 2010 posterior distributions of Bank and RF distributions to forward simulate observationally-derived total emissions for 2011–2016, Bank emissions, and their difference, which we refer to as Direct Total Emissions. The assumption that atmospheric lifetimes are relatively constant from 2010 onwards is broadly supported by ensemble lifetime estimates from the Whole Atmosphere Chemistry Climate Model (WACCM) simulations, shown in Supplementary Fig. 5. This is done for inferred parameter distributions allowing both an unexpected emissions scenario (as in Lickley et al., 2020[6]) and reported emissions scenario in order to provide our best estimate of the sources and magnitude of emissions that can include unexpected production and release if suggested by all the inputs. However, for our post-2010 estimated values, we assume that RF remains constant and we do not account for a portion of further new production going into banks (since this is both poorly known and is expected to be relatively small compared to the existing bank). The assumption of a constant RF post-2010 reflects the posterior RF timeseries derived from the bottom-up accounting model that accounts for changes in bank composition over time (see Supplementary Fig. 6). Results are shown in Fig. 5 for the unexpected emission scenario and the reported production scenario is provided in Supplementary Fig. 7. Direct total emissions are shown in Fig. 6.

## Discussion

Table 2 summarizes the key findings of our study. In this analysis we set out to estimate lifetimes and their uncertainties for CFC-11, 12, and 113, and provide information on their emissions and

sources of emissions. Our analysis suggests that the lifetime of CFC-11 is 49.1 years with a 95% CI of (44.1, 54.6), in the range of most previously estimated values. We estimate CFC-12 and CFC-113 lifetimes to be 84.7 years with a 95% CI of (76.7, 95.5), and 79.9 years with a 95% CI of (74.0, 88.7), respectively. Both mean values for CFC-12 and 113 estimated lifetimes are lower than previous surface mole fraction trend-based estimates[10], although they are statistically consistent, at the 2-sigma level, with SPARC 2013 possible ranges[7]. We attribute this difference in estimated values to the inclusion of a wider range of constraints in our analysis, namely the inclusion of co-dependencies across molecules' lifetimes. This is an important distinction between this work and previous inverse-modeling studies, which did not account for such co-dependencies across molecules. The present study probabilistically accounts for the correlation in lifetimes across gases as exhibited by chemistry-climate models (from SPARC). That is, shorter (longer) than average lifetimes for one gas will likely imply shorter (longer) than average lifetimes for the other gases. Here we show that including such dependencies clarifies our conclusions about lifetimes, as illustrated in Fig. 4 versus Supplementary Fig. 2.

Montzka et al. (2018)[4] used a single set of NOAA observations along with a constant lifetime of 57.5 years, derived from the two 3-D CCM simulations used in their analysis, to estimate total CFC-11 emissions from 2002–2012 to be 54 Gg/yr, and from 2014–2016 to be 67 Gg/yr, with an unexpected increase of 13 ± 5 Gg/yr. Here we use both the AGAGE and NOAA global observational datasets that are merged along with other constraints to estimate average total CFC-11 emissions from 2002–2012 of ~66.6 Gg/yr and from 2014–2016 of ~77.6 Gg/yr, with a difference of 10.9 Gg/yr, and a 95% CI of (10.1, 11.8). Rigby et al. (2019) provided evidence that an emission increase of 7 ± 3 Gg/yr occurred from eastern China between the periods 2008–2012 and 2014–2017. Note that Rigby et al. (2019) use the SPARC recommended lifetime values (e.g., 52 years for CFC-11), thus their emissions estimates represent a lifetime-corrected version of Montzka et al. (2018) values. We find that while a large fraction of total emissions over these time periods do come from banks, significant decreases in bank emissions as banks are depleted (see Table 2) now suggests that a significant increase of direct total emissions is occurring over time. Our best estimate of total direct emissions of CFC-11 from 2014–2016 is 23.2 Gg/yr, with a 95% CI of (13.7, 35.8). Therefore, we find that total unexpected emissions due to new production in breach of the Protocol are likely to be higher than the increase inferred by Montzka et al. (2018)[4] by about 10 Gg/yr. Further, our best estimate exceeds the contribution estimated for eastern China by Rigby et al. (2019) by about 16 Gg/year, supporting the proposal

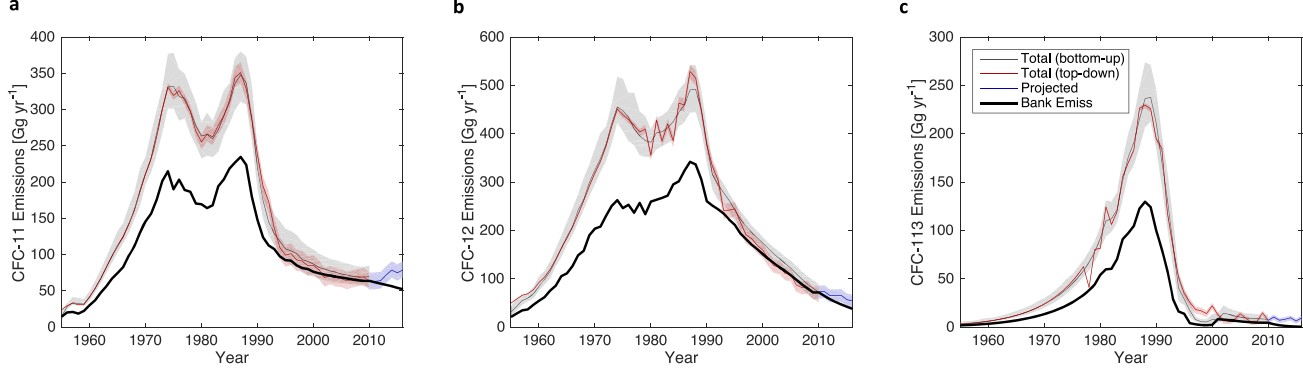

**Fig. 5 Posterior bank and total emissions.** Posterior emissions (red and grey), median posterior bank emissions (black), observationally derived emissions with posterior lifetimes (blue) for (**a**) CFC-11 (**b**) CFC-12, and (**c**) CFC-113. Shaded regions indicate the 95% confidence interval.

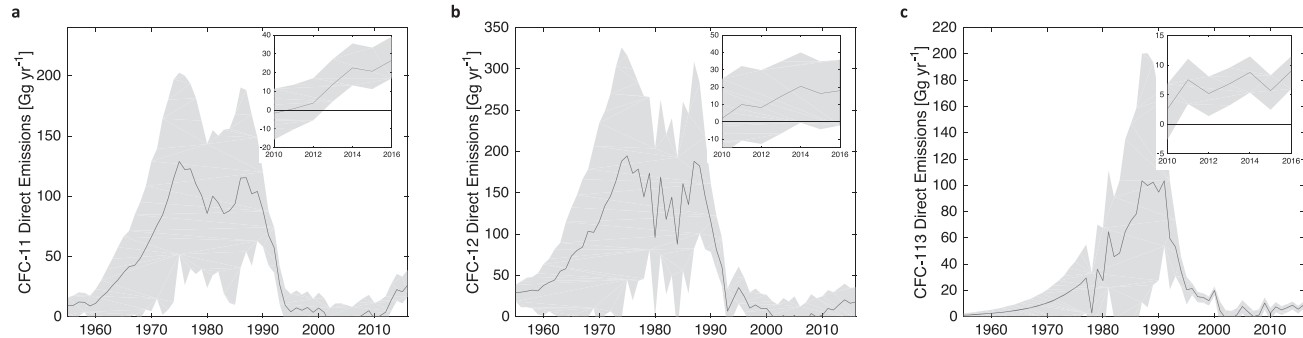

**Fig. 6 Direct total emissions.** Posterior top-down total emissions estimates minus posterior bank emissions for (**a**) CFC-11 (**b**) CFC-12, and (**c**) CFC-113. Shaded region indicates the 95% confidence interval.

| Table 2 Median emissions estimate (95% CI) by source for the unexpected emission scenario. Values are reported in [Gg/yr]. | | | |
|---|---|---|---|
| | **CFC-11** | **CFC-12** | **CFC-113** |
| Bank Emissions 2002–2012 | 66.2 (55.7, 80.0) | 92.3 (71.8, 107.7) | 5.4 (2.0, 9.4) |
| Total Emissions 2002–2012 | 66.6 (55.2, 79.4) | 91.6 (77.4, 104.7) | 8.6 (5.7, 11.0) |
| Direct Total Emissions 2002–2012 | −1.0 (−11.8, 11.8) | 0.0 (−17.9, 22.4) | 3.3 (−1.3, 7.8) |
| Bank Emissions 2014–2016 | 54.4 (47.1, 61.9) | 42.7 (24.9, 58.9) | 0.72 (0.08, 2.8) |
| Total Emissions 2014–2016 | 77.6 (67.0, 89.5) | 59.3 (45.7, 71.8) | 8.6 (5.9, 10.9) |
| Direct Total Emissions 2014–2016 | 23.2 (13.7, 35.8) | 18.3 (−2.3, 36.8) | 7.8 (4.6, 10.4) |

that unexpected sources elsewhere are likely (notwithstanding that Rigby et al., 2019 only estimated the magnitude of the rise in emissions, rather than the excess over a declining bank, as we have in this work). However, the lower limit of our 95% CI is within Rigby et al. (2019)'s stated uncertainty range, indicating that it is unlikely but plausible that eastern China is the only source.

For CFC-12, emissions are consistently decreasing post-2000. Nonetheless, since our estimate of bank emissions are on average decreasing faster than total emissions, significant direct total emissions from 2014–2016 are also very likely to be occurring for this gas. CFC-12 direct emissions for 2014–2016 are estimated to be 18.3 Gg/yr, with uncertainties large enough to contain zero only at the 95% CI. Further, between 2002–2012 and 2014–2016 there is a significant increase in estimated direct emissions of 18.0 Gg/yr, with a 95% CI of (11.8, 22.7); i.e., exceedingly unlikely to include zero. This finding strongly merits further investigation, as CFC-12 production is expected to accompany CFC-11 production in most chemical plants[14]. Estimated total CFC-113 emissions are ~8.6 Gg/yr with an ~95% CI of (6.0, 11.0) for both 2002–2012 and 2014–2016. Further, given the decrease in bank

emissions, we estimate the direct total emissions to have increased to 7.8 Gg/yr in 2014–2016 from 3.3 Gg/yr in 2002–2012, substantially larger than expected from its allowed reported global use in feedstocks (see Lickley et al., 2020). We note that the NOAA and AGAGE data do not separate measurements of CFC-113 and CFC-113a. Recent studies have noted an increase in CFC-113a concentrations[15,16]. Accounting for this trend in CFC-113a and assuming the instruments are equally sensitive to both CFC-113 and CFC-113a, we find direct total emissions of CFC-113 to be 6.4 Gg/yr in 2014–2016, which is within the 95% uncertainty range of our values reported here (see Supplementary Fig. 11).

Overall, our findings strengthen the understanding of estimates and their uncertainties for lifetimes, banks, and apparent direct emissions of all three of the major CFCs, 11, 12, and 113. We have shown that all three molecules are likely being directly emitted in comparable amounts between 2014 and 2016, in contrast to expectations under the Montreal Protocol. Determining the sources of these emissions and whether and how to reduce them is a pressing challenge for the Parties to the Montreal Protocol.

## Methods

**Bayesian parameter estimation modeling framework.** In this section, we discuss the adaptations to the BPE model developed in Lickley et al. (2020). This model is implemented by first developing a bottom-up simulation model for CFC-11, 12, and 113 that uses a collection of input parameters to recursively simulate outputs, which include mole fractions, $M_{j,t}$, emissions, $E_{j,t}$, and banks, $B_{j,t}$, for molecule $j$, in year, $t$. The input parameters include atmospheric lifetimes, $\tau_{j,t}$, production, $Prod_{j,t}$, direct emissions, $DE_{j,t}$, and bank release fractions $RF_{j,t}$. The equations for the deterministic simulation model are shown in Eqs. (1)–(3). Mole fractions are modeled as;

$$M_{j,t} = M_{j,t-1} \times \exp\left(\frac{1}{\tau_{j,t}}\right) + A \times E_{j,t} \qquad (1)$$

where $A$ is a constant that converts units of emissions to units of mole fractions. We follow Daniel et al. (2007)[17] and let $A$ include a fixed factor of 1.07 to account for the increased mixing ratios at the surface relative to globally average values. Emissions, $E_{j,t}$, in Eq. (1) are modeled as;

$$E_{j,t} = RF_{j,t} \times B_{j,t-1} + DE_{j,t} \times Prod_{j,t} \qquad (2)$$

and banks, $B_{j,t}$, in Eq. (2) are modeled recursively as

$$B_{j,t} = (1 - RF_{j,t}) \times B_{j,t-1} + (1 - DE_{j,t}) \times Prod_{j,t} \qquad (3)$$

Prior distributions for each of the input parameters are described below and in Lickley et al. (2020). While the deterministic simulation model is independent across molecules, chemistry-climate models demonstrate that the lifetimes of molecules exhibit interdependencies across molecules. We, therefore, obtain joint posterior distributions of the vector of input parameters by implementing Bayes' theorem as follows;

$$P(\theta_{11}, \theta_{12}, \theta_{113} | \mathbf{D}_{11}, \mathbf{D}_{12}, \mathbf{D}_{113})$$
$$\propto P(\theta_{11}, \theta_{12}, \theta_{113}) P(\mathbf{D}_{11}, |, \theta_{11}, \theta_{12}, \theta_{113}) P(\mathbf{D}_{12}, |, \theta_{11}, \theta_{12}, \theta_{113}) P(\mathbf{D}_{113}, |, \theta_{11}, \theta_{12}, \theta_{113}) \qquad (4)$$

where $\theta_j$'s denote the vector of inputs and outputs of the deterministic simulation model (Eqs. 1–3) and $\mathbf{D}_j$ denotes the data (i.e., observed mole fractions) for molecule, $j$. We assume that the data ($\mathbf{D}_{11}$, $\mathbf{D}_{12}$, and $\mathbf{D}_{113}$) are conditionally independent given $\theta_{11}$, $\theta_{12}$, and $\theta_{113}$. This assumption implies that the relationship between $\mathbf{D}_{11}$, $\mathbf{D}_{12}$, and $\mathbf{D}_{113}$ is captured in the simulation model described in (1–3), but that the errors between data and model are uncorrelated across molecules.

In Eq. (4), $P(\theta_{11}, \theta_{12}, \theta_{113})$ describes the joint prior distribution of the input parameters and outputs for CFC-11, 12, and 113 for all model years. $P(\mathbf{D}_j | \theta_{11}, \theta_{12}, \theta_{113})$ is the multivariate likelihood of all years of observed mole fractions of molecule $j$, given the parameters of the deterministic simulation model as described in Eqs. (1)–(3), i.e., the error between data and simulation model. For computational efficiency, we solve Eq. (4) using sequential Bayesian updating. To do so, we first solve for the posterior $P(\theta_{11}, \theta_{12}, \theta_{113} | \mathbf{D}_{11}, \mathbf{D}_{12})$ where

$$P(\theta_{11}, \theta_{12}, \theta_{113} | \mathbf{D}_{11}, \mathbf{D}_{12})$$
$$\propto P(\theta_{11}, \theta_{12}, \theta_{113}) P(\mathbf{D}_{11} | \theta_{11}, \theta_{12}, \theta_{113}) P(\mathbf{D}_{12} | \theta_{11}, \theta_{12}, \theta_{113}) \qquad (5)$$

This posterior is then used as the prior distribution and updated given observed mole fractions of CFC-113 ($\mathbf{D}_{113}$), obtaining a posterior distribution equivalent to Eq. (4);

$$P(\theta_{11}, \theta_{12}, \theta_{113} | \mathbf{D}_{11}, \mathbf{D}_{12}, \mathbf{D}_{113}) \propto P(\theta_{11}, \theta_{12}, \theta_{113} | \mathbf{D}_{11}, \mathbf{D}_{12}) P(\mathbf{D}_{113} | \theta_{11}, \theta_{12}, \theta_{113}) \qquad (6)$$

The posteriors in Eqs. (4)–(6) are obtained by multiplying by a normalizing constant such that the right-hand side integrates to 1.

To solve for the posterior in Eq. (4), the model is implemented as follows.

1. First develop prior distributions for all input parameters in $\theta$. This includes joint prior distributions for $\tau_{11}$, $\tau_{12}$, and $\tau_{113}$ as well as $Prod_{j,t}$, $DE_{j,t}$, and $RF_{j,t}$.
2. Using Monte Carlo simulation, sample from the prior distributions and simulate many realizations of mole fraction time series using Eqs. (1), (2) and (3).
3. Specify likelihood function of observed mole fractions given simulated mole fractions (i.e., the error between data and model).
4. Estimate the joint posteriors of Eqs. (5) and (6) using the sampling importance ratio method described further below.

We describe each step of the model in more detail below.

**Release fraction and direct emissions priors.** For each molecule, priors are developed for release fraction (RF) and direct emissions (DE) using industry-reported data and previously published emissions functions[12]. As in Lickley et al. (2020), we jointly estimate RF and DE priors using a bottom-up accounting method that evaluates the composition of the bank over time. This includes accounting for various equipment types and release fractions. RF is modeled as a fraction of the bank and depends on the type of equipment in the bank in any given year. DE is modeled as a fraction of production and depends on the type of

equipment produced in a given year. An update since Lickley et al. (2020), is that the prior DE of foam now follows a beta distribution (with parameters 5, 5) centered at 1.5%, based on values reported in Ashford (2004). For a detailed description of other RF and DE priors, see the Supplementary Information in Lickley et al. (2020).

**Production priors.** Production priors are modeled as in Lickley et al. (2020), where a lognormal distribution is assumed with a lower bound of 95% of global reported values. Reported values of global production come from the Alternative Fluorocarbon Environmental Acceptability Study (AFEAS) for years prior to 1989, and from the United Nations Environmental Program (UNEP) from 1989 onwards. We adopt a correction for the AFEAS data following WMO (2002)[11] where AFEAS production values are augmented with production data from UNEP. We assume autocorrelation over time in the bias of reported production and infer this autocorrelation parameter (see Lickley et al. (2020) for more details). For CFC-113, we adjusted prior distributions so that observed mole fractions are contained in the prior simulated mole fraction range. This was achieved by setting a lower bound of production to 70% of reported and doubling the uncertainty range (i.e., changing $B$ from 0.2 to 0.4, see Lickley et al. (2020) for details). We include two scenarios for production, one where the priors are based on reported production and one where both CFC-11 and CFC-113 have "unexpected emissions" from 2000 onwards in line with estimates from Montzka et al. (2018) and Lickley et al. (2020). Since CFC-11, 12, and 113 are considered jointly, the method can also reveal whether unexpected production of CFC-12 is likely.

**Lifetime Priors.** We develop joint priors for the atmospheric lifetimes that take into account the correlation in lifetimes over time and across molecules. We develop priors for lifetimes from the time varying SPARC CCM values as follows:

1. For each of the SPARC CCMs, compute the inverse of each modeled lifetime (since the inverse of lifetimes typically assumed to be normally distributed, Chapter 6 of SPARC[7]). We then smooth the inverse lifetime using a 10-year moving average. Most modeling groups that contributed lifetime estimates to SPARC, considered the period from 1960 to 2010. Some available modeled lifetime calculations end before 2010, in which case we assume the last 10-year averaged value holds for the remainder of the time period.
2. For each molecule, $j$, create a $N_{yrs} \times N_{models}$ matrix, $\mathbf{K}_j$, of the smoothed inverse lifetimes. $N_{yrs}$ is equal to 42; the number of SPARC modeled years after smoothing with a 10-yr window. $N_{models}$ is the number of SPARC models and is equal to 7.
3. We assume that the inverse of these modeled lifetimes are normally distributed, Our prior lifetime distributions are then modeled as;

$$\begin{array}{l} 1/\tau_{11} \\ 1/\tau_{12} \\ 1/\tau_{113} \end{array} \sim N\left( \begin{array}{l} \mu_{11} \\ \mu_{12} \\ \mu_{113} \end{array}, \Sigma^\tau \right) \qquad (7)$$

where $\mu_j$ is a $N_{yrs} \times 1$ vector equal to the row average of $\mathbf{K}_j$ and $\Sigma^\tau$, is a $3N_{yrs} \times 3N_{yrs}$ matrix equal to the covariance of $\begin{bmatrix} \mathbf{K}_{11} \\ \mathbf{K}_{12} \\ \mathbf{K}_{113} \end{bmatrix}$.

4. Samples from this prior will represent the inverse lifetimes from 1965 to 2006. Lifetime samples are extended on each end of the time series to obtain an estimated prior from 1955 to 2010 (i.e., the 1965 value in the sample is applied to all years prior to 1965, and the 2006 value is applied to all subsequent years).

**Likelihood of observed data given modeled mole fractions.** Observed mole fractions come from the merged dataset based on AGAGE and NOAA global mean surface station measurements[18]. Data are available from 1980 to 2018. Due to large uncertainties in unreported emissions since 2012 and due to SPARC time-varying CCM lifetime estimates ending in 2010, we base the likelihood of each simulation on yearly observed data between 1980 and 2010. We assume that;

$$\mathbf{D}_j = \mathbf{M}_j + \epsilon_j, \qquad (8)$$

where $\mathbf{D}_j$ is a $N_{Obs} \times 1$ vector where $N_{Obs}$ corresponds to 31 years and each input is yearly observed mole fractions for molecule, $j$, between 1980 and 2010. $\mathbf{M}_j$ is a $N_{Obs} \times 1$ vector of simulated mole fractions, accounting for dependence across all molecules, and $\epsilon_j$ is a $N_{Obs} \times 1$ vector corresponding to the error term and is assumed to be normally distributed with mean zero and covariance $\Sigma_j^{LF}$. Note that the simulation model described in Eqs. (1)–(3) is for the period 1960–2010 ($N_{yrs} = 51$), and the posterior is conditioned on observations which are available for 1980–2010, hence $N_{Obs} = 31$. The likelihood function is, therefore, a multivariate function of the difference between modeled and

observed mole fractions;

$$P(\mathbf{D_j}, |, \boldsymbol{\theta}_{11}, \boldsymbol{\theta}_{12}, \boldsymbol{\theta}_{113}) = \frac{1}{\sqrt{(2\pi)^{N_{Obs}}|\boldsymbol{\Sigma}_j^{LF}|}} \exp\left(-\frac{1}{2}\boldsymbol{\epsilon}_j^{\mathrm{T}}\left(\boldsymbol{\Sigma}_j^{LF}\right)^{-1}\boldsymbol{\epsilon}_j\right) \quad (9)$$

The covariance matrix, $\boldsymbol{\Sigma}_j^{LF}$, represents the sum of the uncertainties of modeled and observed mole fractions. CFC-11 and 12 are both measured to an estimated accuracy of around 1%, and CFC-113 is measured to an accuracy of ~1.5%[18]. We adopt these values and assume that errors from observations are equivalent to ~1% of observed CFC-11 and CFC-12 mole fractions, and 1.5% of observed CFC-113 mole fractions. We do not know the uncertainty of the simulation model, and chose uncertainties for the simulation model component of uncertainty equivalent to 2% of observed mole fractions for CFC-11 and 12, and 4% of observed mole fractions for CFC-113. Taken together and assuming an additive error model, these choices lead to $\boldsymbol{\Sigma}_j^{LF}$ having diagonal elements equal to $0.03 \times \mathbf{D_j}$ for CFC-11 and CFC-12, and $0.055 \times \mathbf{D_j}$ for CFC-113. We assume high autocorrelation in error terms and include autocorrelations in the off-diagonals of 0.99, 0.99, and 0.98 for CFC-11, CFC-12, and CFC-113, respectively. This parameterizes the strong influence of near-fully correlated uncertainties such as scale errors. The selections for uncertainties and autocorrelation values were based on expert judgment. Autocorrelation values as low as 0.95, and uncertainties ranging from 1–5.5% were tested and results were within uncertainty of those provided here.

**Estimating posteriors**. As in Lickley et al. (2020), the posteriors are estimated using the sampling importance ratio (SIR) method, which involves sampling from the priors and then resampling the prior samples based on an importance ratio proportional to the likelihood function (see Bates et al. (2003)[19] and Hong et al. (2005)[20] for more details). Using a sequential updating approach, we implement the sampling procedure twice, first to obtain the posterior $P(\boldsymbol{\theta}_{11}, \boldsymbol{\theta}_{12}, \boldsymbol{\theta}_{113}|\mathbf{D}_{11}, \mathbf{D}_{12})$ as shown in Eq. (5). For this iteration of the SIR method, the prior distribution is the joint prior distribution of all three molecules, $P(\boldsymbol{\theta}_{11}, \boldsymbol{\theta}_{12}, \boldsymbol{\theta}_{113})$, as described above. The posterior is then obtained by resampling based on weights proportional to the importance ratio;

$$\frac{P(\boldsymbol{\theta}_{11}, \boldsymbol{\theta}_{12}, \boldsymbol{\theta}_{113}|\mathbf{D}_{11}, \mathbf{D}_{12})}{P(\boldsymbol{\theta}_{11}, \boldsymbol{\theta}_{12}, \boldsymbol{\theta}_{113})} \propto P(\mathbf{D}_{11}|\boldsymbol{\theta}_{11}, \boldsymbol{\theta}_{12}, \boldsymbol{\theta}_{113})P(\mathbf{D}_{12}|\boldsymbol{\theta}_{11}, \boldsymbol{\theta}_{12}, \boldsymbol{\theta}_{113}) \quad (10)$$

The posterior distributions obtained in the first iteration are then treated as the priors in the second updating stage. Note that the only component of $\boldsymbol{\theta}_{113}$ that is conditioned on $\mathbf{D}_{11}, \mathbf{D}_{12}$ are the atmospheric lifetimes. To obtain the full posterior, we resample from the updated priors based on weights proportional to the importance ratio;

$$\frac{P(\boldsymbol{\theta}_{11}, \boldsymbol{\theta}_{12}, \boldsymbol{\theta}_{113}|\mathbf{D}_{11}, \mathbf{D}_{12}, \mathbf{D}_{113})}{P(\boldsymbol{\theta}_{11}, \boldsymbol{\theta}_{12}, \boldsymbol{\theta}_{113}|\mathbf{D}_{11}, \mathbf{D}_{12})} \propto P(\mathbf{D}_{113}|\boldsymbol{\theta}_{11}, \boldsymbol{\theta}_{12}, \boldsymbol{\theta}_{113}) \quad (11)$$

To sample from the priors, we use a sample size of $N = 1{,}000{,}000$. The first iteration of resampling uses an $N = 300{,}000$, and the second iteration of resampling uses an $N = 100{,}000$. To check the convergence of the SIR algorithm on the true posterior, the BPE was implemented ten times. The range in estimated values of the median of each lifetime distribution, as well as the 95 and 68% confidence intervals, was within 3% across the 10 iterations of the model, and median values were well within the uncertainty of lifetime estimates reported below.

## Data availability
The datasets generated and/or analyzed during the current study are available at https://github.com/meglickley/CFCLifetimes.

## Code availability
All code used in this work is available at https://github.com/meglickley/CFCLifetimes [21]. All analyses were done in MATLAB.

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

## Acknowledgements

M.L. and S.S. acknowledge the support of VoLo foundation. M.L. acknowledges support from the Martin fellowship and S.S. acknowledges support from the Martin chair.

## Author contributions

M.L., S.F., M.R. and S.S. conceptualized the work. M.L. conducted the analysis. M.L., S.F., M.R. and S.S. contributed to the interpretation of the data. M.L. drafted the paper and M.L., S.F., M.R. and S.S. contributed substantial revisions of the paper.

## Competing interests
The authors declare no competing interests.
