## [Peer Review File · Nature Communications]

Reviewers' Comments:

Reviewer #1:

Remarks to the Author:

In the current version of the manuscript, Lickley et al. have updated the method from a Bayesian framework based on their recent published work (Lickley et al, Nat. Comm., 2020) to infer the lifetimes, banks and emissions of CFC-11, CFC-12 and CFC113. Their probability BPE model requires the input parameters including the atmospheric lifetimes, production, direct emissions, and bank release fractions. This would then derive the modelled bank, Emissions and mole fractions through very complicated and sophisticated procedures which have been described well in the Method section. They have quantified and estimated the emissions during 2014-2016 were 19.6, 16.2, and 7.7 Gg/yr for CFC-11, 12 and 113, respectively.

This is still a hot and active research topic after Montzka et al. (2018) first reported the unexpected increase in CFC11 emissions. One of the other main messages in this paper is how important CFC lifetimes used in the Bayesian Parameter Estimation model (Lickley et al., 2020) to simultaneously get the time series of their banks, emissions and mole fractions.

It is clearly shown in the paper there have some improvements in the Figure 3 when jointly inferred lifetimes of CFC11, 12 and 113 used from SPARC CCMs rather than assuming their lifetimes individually in the BPE model.

Based on this, the authors relied on the values of the posterior lifetime, bank and RF of year 2010 to predict the total emission, Bank emissions and Direct Total Emissions for 2011-2016 and compared the CFC11 emissions with previous published studies. Overall, the paper is well written and the result is sound and interesting. The only concern from me is that the authors chose 2010 posterior lifetime distributions, banks and constant RF to predict post 2010 estimations though I understand this limitation mainly came from the SPARC CCMs simulations.

I would image that the estimated values of CFC emission would change if the authors use the updated CCMs simulations which can be extended to 2020.

It looks that the new estimation numbers in the paper which are still in the range of published literatures.

The other important point is missing in the current manuscript is what the implication is it from this work.

Where are the new sources of these unidentified emissions came from which have not explained by the authors.

Reviewer #2:

Remarks to the Author:

Review for article by Lickley et al., submitted to Nature Communication.

General comments

The authors presents new estimations of atmospheric lifetimes for the three major CFCs in the atmosphere, CFC-11, CFC-12 and CFC-113, together with new estimation of bank emissions and direct emissions. These new data are obtained using a Bayesian framework. The new and innovative additions in the modelling work are 1) to treat molecule lifetimes for the three CFCs as a prior, therefore producing a posterior estimate as well, and 2) to include correlation of lifetime estimation between the three CFCs. The authors underline how different the results of lifetime, bank emissions and direct emissions are when taking into account this correlation, which in itself is a very important result to better understand the current limitations in Bayesian reconstruction of emissions and calls for substantial improvements. In addition, based on their refined modelling framework, the authors

reconstruct unexpected emissions of CFC-11, with values at the higher end or even surpassing estimations by Montzka et al., 2018 and Rigby et al., 2019, but also unexpected emissions for CFC-12 and CFC-113. Such findings are clearly of interest for the parties of the Montreal Protocol.

The article is extremely well constructed. The work progression is very well explained. Figures are well chosen and constructed. The article is straightforward to follow. Implications of model constructions are very well presented and explained. The conclusions about CFC-11 and CFC-12 and their implications are clearly presented and convincing. I believe the Supplement contains sufficient additional information to follow the steps of the modelling work. In short, I support publication in Nature Communication of the manuscript by Lickley et al., after addressing the minor comment below.

My only minor concern is: what about CFC-113, in particular, what about CFC-113a? Until the publication of a CFC-113a record by Laube et al., 2014, later extended by Adcock et al., 2018, the atmospheric history of CFC-113a was unknown. In the Ozone Assessment Report 2018, p. 1.9, the legend of Figure 1-1 states that "NOAA and AGAGE CFC-113 data likely represent some combination of CFC-113 and CFC-113a (although the influence of CFC-113a on NOAA and AGAGE measurements of CFC-113 is likely small), whereas UEA measures CFC-113 and CFC-113a separately (Adcock et al., 2018)".

Based on the dataset from Adcock et al., 2018, the CFC-113a molar fraction represents around 1% of the CFC-113 molar fraction. Reconstructed CFC-113a emissions increased quite abruptly from 2010, reaching 1.7 Gg yr⁻¹ in 2017. We could therefore assume that for most of the reconstruction presented here, this is likely within the uncertainty of the model. However, since the molar fraction of CFC-113a is currently increasing in the atmosphere, it would be good to mention this gas in this paper. In particular, can part of the (CFC-113 + CFC-113a) observed increase be explained by the CFC-113a increase, or is the CFC-113a increase clearly not sufficient to explain the (CFC-113 + CFC-113a) increase?

Minor technical comments

Figure 2, y-axis unit: it is recommended to use SI based units. "Mole fraction" is correct, and the corresponding SI-based unit is pmol mol⁻¹, for picomole per mole (picomole of the given substance per mole of dry air). "ppt" stands for part per trillion, which is widely used, but unfortunately does not contain the information that the quantity in question is a fraction of moles.

Additional references

Adcock, K. E., Reeves, C. E., Gooch, L. J., Leedham Elvidge, E. C., Ashfold, M. J., Brenninkmeijer, C. A. M., Chou, C., Fraser, P. J., Langenfelds, R. L., Mohd Hanif, N., O'Doherty, S., Oram, D. E., Ou-Yang, C.-F., Phang, S. M., Samah, A. A., Röckmann, T., Sturges, W. T., and Laube, J. C.: Continued increase of CFC-113a (CCl₃CF₃) mixing ratios in the global atmosphere: emissions, occurrence and potential sources, *Atmos. Chem. Phys.*, 18, 4737–4751, <https://doi.org/10.5194/acp-18-4737-2018>, 2018.

Laube, J., Newland, M., Hogan, C. et al. Newly detected ozone-depleting substances in the atmosphere. *Nature Geosci* 7, 266–269 (2014). <https://doi.org/10.1038/ngeo2109>.

Reviewer #3:

Remarks to the Author:

This is an interesting and important paper on a very topical subject. I recommend publication subject to addressing my comments below.

1) A major general comment is on the presentation of lifetimes. I find that the paper makes no distinction between uncertainties in the lifetimes (due to method used to derive the lifetime or uncertainties in kinetics) versus conceptual differences in what lifetime means. For example, the SPARC time-varying lifetimes are presented in Figure 4 and they show an apparent large decreasing trend. There is no explanation of this for the non-expert reader. My understanding is that this trend, during the period of strongly increasing atmospheric loading of CFCs, is due to the disequilibrium in the atmosphere. Essentially large masses of CFCs are in the troposphere and have not yet reached the stratospheric sink region. Hence there is an apparently longer-than-expected lifetime. Eventually the atmosphere approaches an equilibrium. A 'steady-state' lifetime is a single number and so easier to diagnose (from a model) and discuss. This value should be close the later values of the time-varying lifetime. Moreover, as emissions are now low and CFCs are decaying, the steady-state lifetime should be a fair approximation for the atmosphere going forward. I think that the paper needs to explain these concepts.

This distinction could also be mentioned when referring to Figure 1. The two lifetime assumptions are not just different estimates of numbers, but are capturing (or not) different expected atmospheric behaviour.

2) Also, on the topic of lifetimes I think that some more discussion and clarification of the currently recommended values is important (as a headline conclusion is that currently recommend lifetimes of two of the CFCs may be wrong). The SPARC report gave overall recommendations (given in Table 1) which were based on observationally derived values and models. The model results were available for all species, but the availability of the observation-derived values varied. Looking at Table 6.1 from the SPARC report the model-derived steady-state lifetimes seem to generally match the results from this work better. For CFC-11 the recommended lifetime is similar to the modelled lifetime because the inversion-based lifetime (with a large weighting) is 53 years. The situation is different for CFC-12 and CFC-113 where the recommended lifetime is increased by the large weightings (esp for CFC-113) due to values from the inverse modelling. I think that the paper should give some brief overview of how these recommended SPARC values are a combination of models and observations, especially as the model results themselves are shown in other figures. Also, as this paper has authors expert in inverse modelling, can they suggest why the SPARC inverse modelling value might be more realistic for CFC-11 compared to CFC-12 and CFC-113 (to add to the discussion in paragraph starting on line 282)?

While the model values must be self-consistent between themselves for given photochemical data, there could be a systematic error in that data for one or more of the molecules considered. What would be the impact of that on the analysis?

3) Abstract Line 22. 'We find that lifetimes of all three gases are likely shorter than previous estimates...'. This sentence is too general to be absolutely true. There are some estimates for at least some of the gases which are shorter than the new values (e.g. 45 yrs for CFC-11). Do the authors mean the currently recommended values? Or the values used in the latest WMO report? Be more specific.

4) Line 24. '... faster than inferred emission'. The abstract-only reader may not understand what is meant by inferred emission. You could add in 'total' to make it clear that the bank emissions are a contribution to the total inferred. (Nb the caption of Figure 1 also uses the term 'inferred total emissions').

5) Line 32-33. 'The Montreal Protocol.... 2010'. This line seems too general as it stands and needs to be qualified. The abstract discusses CFCs but now this about all ODSs. Production of some

replacements has continued.

6) Line 77. 'are delayed'.

7) Figure 1. It would be helpful if the panels could have legends which explain the different lines. It is tedious to refer to the caption to get an overview of which line is which. Also, it would be good to give the mean value of the SPARC time-varying lifetime. The text (line 78) says that the figure shows the impact of small differences in assumptions, but the mean SPARC time-varying lifetime is surely quite different (judging from Figure 4)?

8) Figure 4. The lifetimes (esp CFC-11) seem to turn upwards at the end of the record, rather than reach some steady state value. Is that an artefact of the smoothing? Or variability in the CCMs?

In the following we include our inline responses to each comment in red along with line numbers that refer to the lines in the track-changes version of the revised Manuscript.

REVIEWER COMMENTS

Reviewer #1 (Remarks to the Author):

In the current version of the manuscript, Lickley et al. have updated the method from a Bayesian framework based on their recent published work (Lickley et al, Nat. Comm., 2020) to infer the lifetimes, banks and emissions of CFC-11, CFC-12 and CFC113. Their probability BPE model requires the input parameters including the atmospheric lifetimes, production, direct emissions, and bank release fractions. This would then derive the modelled bank, Emissions and mole fractions through very complicated and sophisticated procedures which have been described well in the Method section. They have quantified and estimated the emissions during 2014-2016 were 19.6, 16.2, and 7.7 Gg/yr for CFC-11, 12 and 113, respectively.

This is still a hot and active research topic after Montzka et al. (2018) first reported the unexpected increase in CFC11 emissions. One of the other main messages in this paper is how important CFC lifetimes used in the Bayesian Parameter Estimation model (Lickley et al., 2020) to simultaneously get the time series of their banks, emissions and mole fractions. It is clearly shown in the paper there have some improvements in the Figure 3 when jointly inferred lifetimes of CFC11, 12 and 113 used from SPARC CCMs rather than assuming their lifetimes individually in the BPE model.

Based on this, the authors relied on the values of the posterior lifetime, bank and RF of year 2010 to predict the total emission, Bank emissions and Direct Total Emissions for 2011-2016 and compared the CFC11 emissions with previous published studies. Overall, the paper is well written and the result is sound and interesting.

The only concern from me is that the authors chose 2010 posterior lifetime distributions, banks and constant RF to predict post 2010 estimations though I understand this limitation mainly came from the SPARC CCMs simulations. I would image that the estimated values of CFC emission would change if the authors use the updated CCMs simulations which can be extended to 2020. It looks that the new estimation numbers in the paper which are still in the range of published literatures.

While we agree that using our BPE derived lifetimes from 2010 to infer post-2010 emissions could have limitations (in particular that prior RF and lifetimes are held constant in order to infer total emissions and bank emissions), we argue that there are also some advantages and a rationale to this approach. We think that stopping the BPE model in 2010 provides a well-reasoned approach to banks and lifetimes for a few key reasons. First, it does not require any assumptions about production type or quantity for the priors past 2010. This is important as there is no reported data on the type or quantity of CFC-11 and 12 production beyond 2010 (since it is occurring outside the Montreal Protocol), so by ending the BPE in 2010, we do not risk making an ill-informed guess of the key priors which can influence the results. The second

issue, which is raised by the reviewer, is that the publicly available SPARC models run with emission boundary conditions for that assessment do not extend past 2010 to our knowledge. To address this issue, we did examine simulations from the Whole Atmosphere Chemistry Climate Model (WACCM) lifetimes past 2010. The revised draft includes lifetime results from a 10-member ensemble of simulated lifetimes in the supplement (Fig S5), indicating that WACCM modeled lifetimes are relatively stable after 2010; this supports the assumption in our analysis and thus provides a reasonable top-down estimate of emissions past 2010. While we do not have space to get into this in the paper, the range of those results gives an indication of uncertainty, and the periodic cycle observed in each of the ensemble members is a result of the imposed quasi biennial oscillation. We've added some clarifying text explaining our rationale (Line 305 - 311). We have also explained why lifetimes are relatively constant after 2010, in response to a comment from Reviewer 3 (below); that information complements this discussion.

The assumption that atmospheric lifetimes are relatively constant from 2010 onwards is broadly supported by ensemble lifetime estimates from the Whole Atmosphere Chemistry Climate Model (WACCM) simulations, shown in Figure S5.

Regarding the question of whether a stable RF for banks past 2010 is a reasonable assumption, we note the following: Priors for RF and direct emissions are modeled using a bottom-up simulation model that accounts for equipment type over time, so we are able to provide our best guess of how RF and DE would change given the posterior bank composition in 2010. We have included this additional analysis in the revised supplement (Figure S6). This added figure illustrates that RF is relatively constant for each of CFC-11, 12 and 113, since the composition of the bank does not change substantially over this time period. While it is true that RF could change based on the unexpected new production, we are interested in this paper in quantifying old bank contributions to ongoing emissions, which excludes the new production. We have also amended the main text to note this assumption: Lines 318-320

The assumption of a constant RF post-2010 reflects the posterior RF timeseries derived from the bottom-up accounting model that accounts for changes in bank composition over time (see Figure S6).

The other important point is missing in the current manuscript is what the implication is it from this work. Where are the new sources of these unidentified emissions came from which have not explained by the authors.

Our analysis uses globally aggregated production data by equipment type and globally averaged observed mole fractions. We are not able to identify the geographic location of these emissions with this modeling approach. We agree that this is motivation for further work though beyond the scope of this paper. We cite in the text other work that has identified sources of the new CFC-11 (Line 710-712):

Rigby et al. (2019) provided evidence that an emission increase of 7 ± 3 Gg/yr occurred from eastern China between the periods 2008 – 2012 and 2014 - 2017.

We acknowledge that our work calls for additional analysis to further investigate the source of these new emissions on (Lines 787-788):

Determining the sources of these emissions and whether and how to reduce them is a pressing challenge for the Parties to the Montreal Protocol.

Reviewer #2 (Remarks to the Author):

Review for article by Lickley et al., submitted to Nature Communication.

General comments

The authors presents new estimations of atmospheric lifetimes for the three major CFCs in the atmosphere, CFC-11, CFC-12 and CFC-113, together with new estimation of bank emissions and direct emissions. These new data are obtained using a Bayesian framework. The new and innovative additions in the modelling work are 1) to treat molecule lifetimes for the three CFCs as a prior, therefore producing a posterior estimate as well, and 2) to include correlation of lifetime estimation between the three CFCs. The authors underline how different the results of lifetime, bank emissions and direct emissions are when taking into account this correlation, which in itself in a very important result to better understand the current limitations in Bayesian reconstruction of emissions and calls for substantial improvements. In addition, based on their refined modelling framework, the authors reconstruct unexpected emissions of CFC-11, with values at the higher end or even surpassing estimations by Montzka et al., 2018 and Rigby et al, 2019, but also unexpected emissions for CFC-12 and CFC-113. Such findings are clearly of interest for the parties of the Montreal Protocol.

The article is extremely well constructed. The work progression is very well explained. Figures are well chosen and constructed. The article is straightforward to follow. Implications of model constructions are very well presented and explained. The conclusions about CFC-11 and CFC-12 and their implications are clearly presented and convincing. I believe the Supplement contains sufficient additional information to follow the steps of the modelling work. In short, I support publication in Nature Communication of the manuscript by Lickley et al., after addressing the minor comment below.

Thank you!

My only minor concern is: what about CFC-113, in particular, what about CFC-113a? Until the publication of a CFC-113a record by Laube et al., 2014, later extended by Adcock et al., 2018, the atmospheric history of CFC-113a was unknown. In the Ozone Assessment Report 2018, p. 1.9, the legend of Figure 1-1 states that "NOAA and AGAGE CFC-113 data likely represent some combination of CFC-113 and CFC-113a (although the influence of CFC-113a on NOAA and AGAGE measurements of CFC-113 is likely small), whereas UEA measures CFC-113 and CFC-113a separately (Adcock et al., 2018)".

Based on the dataset from Adcock et al., 2018, the CFC-113a molar fraction represents around 1% of the CFC-113 molar fraction. Reconstructed CFC-113a emissions increased quite abruptly

from 2010, reaching 1.7 Gg yr⁻¹ in 2017. We could therefore assume that for most of the reconstruction presented here, this is likely within the uncertainty of the model. However, since the molar fraction of CFC-113a is currently increasing in the atmosphere, it would be good to mention this gas in this paper. In particular, can part of the (CFC-113 + CFC-113a) observed increase be explained by the CFC-113a increase, or is the CFC-113a increase clearly not sufficient to explain the (CFC-113 + CFC-113a) increase?

Thanks for this helpful comment. We tested our results using the Adcock estimate of increasing CFC-113a concentrations from 0.5 ppt in 2012 to 0.7 ppt in 2017. If these values are removed from the WMO concentrations (implying that CFC-113 and 113a are measured with the same sensitivity in the available global data used), we find that our inferred emissions are still within the same range and remain significantly different from zero, supporting our key point about emissions of this gas. A new figure showing this has been added, Figure S11. We also added explanatory text in the main text, Line 758 - 763:

We note that the NOAA and AGAGE data do not separate measurements of CFC-113 and CFC-113a. Recent studies have noted an increase in CFC-113a concentrations^{15,16}. If we account for this trend in CFC-113a and assume the instruments are equally sensitive to both CFC-113 and CFC-113a, then we find direct total emissions of CFC-113 to be 6.4 Gg/yr in 2014-2016, which is within the 95% uncertainty range of our values reported here (see Figure S10).

Minor technical comments

Figure 2, y-axis unit: it is recommended to use SI based units. "Mole fraction" is correct, and the corresponding SI-based unit is pmol mol⁻¹, for picomole per mole (picomole of the given substance per mole of dry air). "ppt" stands for part per trillion, which is widely used, but unfortunately does not contain the information that the quantity in question is a fraction of moles.

Thank you for this correction, we have changed Figure 2's y-axis accordingly.

Additional references

Adcock, K. E., Reeves, C. E., Gooch, L. J., Leedham Elvidge, E. C., Ashfold, M. J., Brenninkmeijer, C. A. M., Chou, C., Fraser, P. J., Langenfelds, R. L., Mohd Hanif, N., O'Doherty, S., Oram, D. E., Ou-Yang, C.-F., Phang, S. M., Samah, A. A., Röckmann, T., Sturges, W. T., and Laube, J. C.: Continued increase of CFC-113a (CCl₃CF₃) mixing ratios in the global atmosphere: emissions, occurrence and potential sources, *Atmos. Chem. Phys.*, 18, 4737–4751, <https://doi.org/10.5194/acp-18-4737-2018>, 2018.

Laube, J., Newland, M., Hogan, C. et al. Newly detected ozone-depleting substances in the atmosphere. *Nature Geosci* 7, 266–269 (2014). <https://doi.org/10.1038/ngeo2109>.

These references have been added

Reviewer #3 (Remarks to the Author):

This is an interesting and important paper on a very topical subject. I recommend publication

subject to addressing my comments below.

1) A major general comment is on the presentation of lifetimes. I find that the paper makes no distinction between uncertainties in the lifetimes (due to method used to derive the lifetime or uncertainties in kinetics) versus conceptual differences in what lifetime means. For example, the SPARC time-varying lifetimes are presented in Figure 4 and they show an apparent large decreasing trend. There is no explanation of this for the non-expert reader. My understanding is that this trend, during the period of strongly increasing atmospheric loading of CFCs, is due to the disequilibrium in the atmosphere. Essentially large masses of CFCs are in the troposphere and have not yet reached the stratospheric sink region. Hence there is an apparently longer-than-expected lifetime. Eventually the atmosphere approaches an equilibrium. A 'steady-state' lifetime is a single number and so easier to diagnose (from a model) and discuss. This value should be close the later values of the time-varying lifetime. Moreover, as emissions are now low and CFCs are decaying, the steady-state lifetime should be a fair approximation for the atmosphere going forward. I think that the paper needs to explain these concepts.

This distinction could also be mentioned when referring to Figure 1. The two lifetime assumptions are not just different estimates of numbers, but are capturing (or not) different expected atmospheric behaviour.

This is a great point and we agree that some explanatory text is needed. As per the reviewer's recommendation, we've added some text to explain steady-state lifetime versus the time varying lifetime resulting from the disequilibrium in the atmosphere. Lines 92-97:

It takes approximately 3-7 years for air to be transported from the source of emission to the upper atmospheric loss region and back to the troposphere. SPARC models thus predict a decreasing trend in lifetimes, reflecting the disequilibrium between tropospheric and stratospheric concentrations when tropospheric concentrations are increasing. A near steady-state lifetime is achieved a few years after tropospheric concentrations stop their rapid rise.

2) Also, on the topic of lifetimes I think that some more discussion and clarification of the currently recommended values is important (as a headline conclusion is that currently recommend lifetimes of two of the CFCs may be wrong). The SPARC report gave overall recommendations (given in Table 1) which were based on observationally derived values and models. The model results were available for all species, but the availability of the observation-derived values varied. Looking at Table 6.1 from the SPARC report the model-derived steady-state lifetimes seem to generally match the results from this work better. For CFC-11 the recommended lifetime is similar to the modelled lifetime because the inversion-based lifetime (with a large weighting) is 53 years. The situation is different for CFC-12 and CFC-113 where the recommended lifetime is increased by the large weightings (esp for CFC-113) due to values from the inverse modelling. I think that the paper should give some brief overview of how these recommended SPARC values are a combination of models and observations, especially as the

model results themselves are shown in other figures. Also, as this paper has authors expert in inverse modelling, can they suggest why the SPARC inverse modelling value might be more realistic for CFC-11 compared to CFC-12 and CFC-113 (to add to the discussion in paragraph starting on line 282)?

We agree that this is a key point in our paper that must be stressed. In particular, the primary limitation in the SPARC recommended values is their assumptions about emissions. If these assumptions are biased, then this will bias lifetime estimates. Along with our explanation in the introduction, we've added to the discussion section some text to explain why our estimates would differ from the SPARC recommended estimates. Lines 294-302:

Why would our CFC-11 lifetime estimate agree with SPARC recommended lifetimes, but not CFC-12 and 113? We attribute this to two factors. The first is that the SPARC recommended values do not account for the inter molecule correlations exhibited by the CCM modeled lifetimes. The second (which is a key limitation in earlier studies including SPARC) is that the recommended lifetimes depend on emissions assumptions, whereas our analysis jointly infers emissions along with lifetimes. We expect that the SPARC prescribed emissions for CFC-12 and 113 are biased low relative to CFC-11, leading to this difference in recommended values.

While the model values must be self-consistent between themselves for given photochemical data, there could be a systematic error in that data for one or more of the molecules considered. What would be the impact of that on the analysis?

This is an interesting point. If there was a systematic error in the data then we believe this would be accounted for by the correlation parameter used to construct the likelihood function. I.e. the off-diagonals in the covariance matrix in equation 9 accounts for high temporal correlation in the residuals (observed mole fractions - modeled mole fractions).

3) Abstract Line 22. 'We find that lifetimes of all three gases are likely shorter than previous estimates...'. This sentence is too general to be absolutely true. There are some estimates for at least some of the gases which are shorter than the new values (e.g. 45 yrs for CFC-11). Do the authors mean the currently recommended values? Or the values used in the latest WMO report? Be more specific.

This is a good point, we changed this to: 'currently assessed values'. The text contains the reference to WMO (2018).

4) Line 24. '... faster than inferred emission'. The abstract-only reader may not understand what is meant by inferred emission. You could add in 'total' to make it clear that the bank emissions are a contribution to the total inferred. (Nb the caption of Figure 1 also uses the term 'inferred total emissions').

This has been changed to 'total emissions'

5) Line 32-33. 'The Montreal Protocol.... 2010'. This line seems too general as it stands and needs to be qualified. The abstract discusses CFCs but now this about all ODSs. Production of some replacements has continued.

Line 36: ODSs has been changed to CFCs.

6) Line 77. 'are delayed'.

Line 91: This has been changed.

7) Figure 1. It would be helpful if the panels could have legends which explain the different lines. It is tedious to refer to the caption to get an overview of which line is which. Also, it would be good to give the mean value of the SPARC time-varying lifetime. The text (line 78) says that the figure shows the impact of small differences in assumptions, but the mean SPARC time-varying lifetime is surely quite different (judging from Figure 4)?

Figure 1 has been updated to add legends in addition to the explanation in the text. We have added the mean of the time-varying SPARC lifetime estimates to the figure caption. We also changed the wording to say instead 'different plausible lifetime and production assumptions'. (Line 92).

8) Figure 4. The lifetimes (esp CFC-11) seem to turn upwards at the end of the record, rather than reach some steady state value. Is that an artefact of the smoothing? Or variability in the CCMs?

Thank you for pointing this out. This is a result of the smoothing/averaging. Some of the CCMs with lower lifetimes ended earlier than others and their end lifetimes weren't carried forward in the averaging. This has been corrected and we have corrected the results throughout accordingly.

Reviewers' Comments:

Reviewer #1:

Remarks to the Author:

The authors have now sufficiently addressed the comments and could be published.

Reviewer #2:

Remarks to the Author:

The authors have well addressed the comments listed in my first review. I now support publication of the revised version of the manuscript in Nature Communication.

Technical comment in the revised version:

- l. 33: "ODS" should be explained.

Reviewer #3:

Remarks to the Author:

The line numbers in this review refer to the PDF of the resubmitted paper with track changes (269803_1_related_ms_5190528_q1p11k.pdf).

As I said in my first review, I find this an interesting and useful paper that I would expect to be published without too much further work. However, I did have some concerns about the discussion of lifetimes, which are a main focus of this paper. I am sorry but I don't find that the edits made in response to my first set of comments have really resolved the issues that I raised. Therefore, I think that more revisions to the text are necessary. My previous comments were quite general because I thought that would be enough to motivate changes throughout the paper. This time I am giving some very specific comments in the hope that this will make the points more helpfully and clearly.

As the paper mentions SPARC a lot, and refers to lifetime estimates used by other studies, I think it is important to clarify some terms and concepts at the start of this review. I think that the authors should use similar specific terms throughout the paper when editing the paper in response to my comments. A lot of my comments refer back to similar lack of clarity over different methods and models.

The SPARC report (SPARC, 2013) provides recommended steady-state lifetimes. The lifetimes presented in Chapter 6 of that reports are the ones typically used in other studies that simply aim to take an optimum single value for a species. Notably, the SPARC recommended lifetimes are used in the WMO/UNEP Assessments (i.e. 2014 and 2018). According to Table 6.1 in SPARC (2013), the recommended steady-state lifetimes are based on 4 methods: (i) forward CCM simulations of steady-state lifetimes, (ii) inverse modelling (IM) of surface observations and assumed fluxes, (iii) tracer-tracer correlations and (iv) satellite burdens and loss rates. The final SPARC recommended value is a weighted mean of these methods, but the weighting varies because not all methods are available for all molecules. Methods (i) and (ii) are relevant to this study and do provide values for all gases. Furthermore, the forward CCMs were also used to derive 'time-varying lifetimes' based on the observed variation of trace gases. As you know these time-varying lifetimes can be substantially different to the steady-state lifetimes, especially when CFCs are growing rapidly.

My specific comments are below. These are arranged in line order (so the simple edits are mixed in with the more important comments).

Specific Comments.

1) Line 22. Change 'assessed' to 'recommended'. The key point is that the SPARC report aimed to provide recommended lifetimes that are then used in e.g. the WMO/UNEP Ozone Assessments. It is not clear what 'assessed' means.

2) Paragraph starting Line 60. This is the main paragraph where the ideas of uncertainty in lifetimes is introduced. It mentions different methods used to infer lifetimes, but does not clarify the 'time-varying' versus 'steady-state' which are used later with the same terminology (see text above). Moreover, it should be stated that it is the forward CCMs which give time time-varying values. There are a few other edits needed in this paragraph:

Line 62. CCM is always chemistry-climate model (not chemical).

Line 66. WCRP is a proper name and officially written as Programme.

Line 66. 'Observations of surface concentrations are generally used only as a qualitative check in such calculations'. This is not true. For transient CCM simulations with surface mixing ratio boundary conditions these values are typically used as input and define the time dependence of the calculations. This was done for the SPARC CCMs for all gases including the three used here. I have looked at the SPARC report (and reference 9) and some flux boundary runs were done for CFC-11 and CFC-12, but those numbers did not feed into the recommended values. For steady-state calculations the surface vmr (or flux) would be irrelevant as the amount of tracer would cancel out in the burden/loss rate calculation.

Line 73. 'The second approach...'. It would be good to give this method a name because limitations of this approach are raised in the discussion. I believe that this is method (ii) in my paragraph above?

Line 75. 'These different approaches have led to wide ranges in estimated lifetimes (see Table 1)': I don't think that this statement matches the previous text in this paragraph on the two methodologies and the idea of time-varying lifetimes. Aside from the new values from the submitted work (first two columns), Table 1 gives previous lifetime estimates from WMO (2003)/IPCC (2001), the SPARC (2013) steady-state estimate and Rigby et al (2013). I have not checked the source of WMO (2003) but I suspect that these are also steady-state values. I assume that Rigby et al (2013) is the Bayesian method? As noted above, the SPARC (2013) steady-state values are a combination of different methodologies.

Line 76: 'therefore large differences in inferred emissions (Chapter 6 of SPARC7)'. I have looked and this Chapter does not give inferred emission. It only presents lifetimes. Need a different reference.

3) Line 92. 'different plausible lifetime'. Figure 1 focuses on the past with large growth in ODSs. The time-varying lifetime will be better suited to this than the steady-state lifetime. I would say that for lifetimes the message from this figure is to use the time-varying ones. If the two different concepts were clear and named then this point could be made.

4) Line 157 'SPARC CCM modelled...' Please add 'time-varying'.

5) Line 162 'set of models'. Change to 'set of CCMs' for clarity.

6) Line 162. 'lower than average CFC-11 lifetimes'. Might want to write this as singular: 'a lower than average CFC-11 lifetime'.

7) Line 178-179. Add words: 'using the SPARC CCM time-varying values'.

8) Line 207 'constant scenario equal to the SPARC multi-model mean'. This is not clear. Do you mean a time-varying multimodel mean (e.g. as shown in Figure 4) or a single fixed value which is the time average of the multimodel mean? The first part of the sentence makes it sound like this constant scenario is not time varying.

9) Line 218/219. 'are within the uncertainty of posterior observationally-derived emissions'. Could add 'our' before 'posterior' to make it clear that you mean your observationally derived emissions. Similarly could add 'our' before 'posterior' in line 221.

10) Line 236/237 'shorter than the SPARC multi-model mean values'. Add 'time-varying'. Could also add 'throughout the time period'.

11) Line 248. Change 'model' to 'CCM'.

12) Line 250 'model'. Please be more specific with which model. 'Bayesian simulation model'? Also (line 251) I don't think DE is defined in the main text so write in full?

13) Line 266. 'The posterior time-averaged lifetime estimates and 2010 lifetime estimates...'. A bit confusing. These are both posterior. Change to e.g. 'The posterior time-averaged and 2010 lifetime estimates....'.

14) Line 266. This is the first mention of the 2010 lifetime specifically in the main text. Please explain why you are showing the 2010 values (readers may not have looked at the Methods).

15) Line 268. 'agrees with SPARC recommended...'. Need to say that this (in Table 1) is the SPARC steady-state lifetime (i.e. the only SPARC recommendation). At this point I think it is important for the readers to know the SPARC recommendation is not simply the SPARC CCM value, but is also based on other methods. I think you should also say here that your 2010 lifetime is the one you would expect to be comparable with the SPARC recommended value based on the methodologies.

16) Line 273. 'The SPARC recommended lifetimes do not account for the intermolecule correlations...'. This is not completely true, but the picture is messy. The steady state CCM values that feed into the recommended values certainly do contain these correlations. The tracer-tracer correlation method (not discussed here) should have this information. The other methods, notably the inversion studies do not.

The relative variation of the new lifetimes derived here match the relative variation of the SPARC CCM values very well. The other methodologies, notably the inverse modelling give the different relative values that feed into the recommended numbers. The situation is more complicated than suggested. This paper is providing improved estimates from inverse modelling but these are now not independent of the CCM values. An improvement to the recommended values would still have to account for the other independent methodologies. It would be better to compare the new numbers with the inverse modelling first rather than discussing solely the SPARC recommended values.

17) Line 275. 'key limitation to in'. Again, this is too general to be completely true. The forward CCM methodology does not depend on the emissions. The inverse modelling does depend on these. So, to be accurate, I think that the statement should be that key limitation applies to the inverse modelling method that feeds into the SPARC recommendation.

18) Line 276. 'the recommend lifetimes depend on emission assumptions'. The recommended lifetimes are based on a combination of sources. As the recommended ones are 'steady state' the emissions are not relevant for the forward running CCMs; the steady-state lifetimes could be inferred with any emission value. For the observationally derived values in the SPARC report whether emissions are relevant will likely depend on the methodology.

19) Line 277-278 'We expect that the SPARC prescribed emissions for CFC-12 and 113 are biased low relative to CFC-11'. Following on from the comments above, I believe that this is only relevant for the inverse modelling contribution to the SPARC recommended lifetimes.

20) Line 301. Table 1. Please standardise the headings for columns 1 and 2. Both should say 'posterior'. State the time span for the average in column 2. Some information could be moved up to a heading which spans both columns so it is clear what the difference is (2010 versus time average).

21) Line 459-460. 'lower than previous surface mole fraction trend-based estimates¹⁰'. Is that really the key point that these lifetimes are based on surface model fractions? They are based on inversions with a fairly simple global model. I think you need to relate this to the 'inverse modelling' methodology and use a consistent name.

22) Line 464 'previous publications'. Specify previous inverse modelling or Bayesian estimation publications. (CCM papers and tracer-tracer correlations would contain this information).

22) Line 509. 'lifetime of 57 years'. I know that it is not your paper (though one co-author overlaps), but having read the detail of lifetimes and got to this point, the reader will wonder why 57 years was used. Why not the SPARC/WMO recommended value? As the Montzka et al study is very high profile, and motivates this work, it would be helpful to say why they used this alternative value. This paragraph also compares the new emissions with those from Rigby et al (2019), but you don't say what lifetime was used there. Was that also 57 years? Please clarify that.

23) Line 689 'SPARC modelled values'. Clarify this is from the SPARC CCMs – e.g. 'from the time varying SPARC CCM values'.

24) Line 690 'Compute the inverse of each modelled lifetime...'. Not really clear that you mean the inverse of each lifetime estimate for each molecule from the separate participating SPARC CCMs. Bullet point 2 gives the number of models but this could be stated higher up in this subsection.

25) Line 718 Add words 'SPARC time-varying CCM lifetime estimates....'

All of the line numbers in the following response are with reference to the pdf of track changes.

REVIEWER COMMENTS

Reviewer #1 (Remarks to the Author):

The authors have now sufficiently addressed the comments and could be published.

Thank you!

Reviewer #2 (Remarks to the Author):

The authors have well addressed the comments listed in my first review. I now support publication of the revised version of the manuscript in Nature Communication.

Thank you!

Technical comment in the revised version:

- l. 33: "ODS" should be explained.

This has been corrected (line 34).

Reviewer #3 (Remarks to the Author):

The line numbers in this review refer to the PDF of the resubmitted paper with track changes (269803_1_related_ms_5190528_qlp11k.pdf).

As I said in my first review, I find this an interesting and useful paper that I would expect to be published without too much further work. However, I did have some concerns about the discussion of lifetimes, which are a main focus of this paper. I am sorry but I don't find that the edits made in response to my first set of comments have really resolved the issues that I raised. Therefore, I think that more revisions to the text are necessary. My previous comments were quite general because I thought that would be enough to motivate changes throughout the paper. This time I am giving some very specific comments in the hope that this will make the points more helpfully and clearly.

Thank you for the more detailed comments, we have done our best to address them below.

As the paper mentions SPARC a lot, and refers to lifetime estimates used by other studies, I think it is important to clarify some terms and concepts at the start of this review. I think that the authors should use similar specific terms throughout the paper when editing the paper in response to my comments. A lot of my comments refer back to similar lack of clarity over different methods and models.

The SPARC report (SPARC, 2013) provides recommended steady-state lifetimes. The lifetimes presented in Chapter 6 of that reports are the ones typically used in other studies that simply aim to take an optimum single value for a species. Notably, the SPARC recommended lifetimes are used in the WMO/UNEP Assessments (i.e. 2014 and 2018). According to Table 6.1 in SPARC (2013), the

recommended steady-state lifetimes are based on 4 methods: (i) forward CCM simulations of steady-state lifetimes, (ii) inverse modelling (IM) of surface observations and assumed fluxes, (iii) tracer-tracer correlations and (iv) satellite burdens and loss rates. The final SPARC recommended value is a weighted mean of these methods, but the weighting varies because not all methods are available for all molecules. Methods (i) and (ii) are relevant to this study and do provide values for all gases. Furthermore, the forward CCMs were also used to derive 'time-varying lifetimes' based on the observed variation of trace gases. As you know these time-varying lifetimes can be substantially different to the steady-state lifetimes, especially when CFCs are growing rapidly.

Thank you for clarifying what is missing here. We've added some more text to clarify what the SPARC recommended values include (see below).

My specific comments are below. These are arranged in line order (so the simple edits are mixed in with the more important comments).

Specific Comments.

1) Line 22. Change 'assessed' to 'recommended'. The key point is that the SPARC report aimed to provide recommended lifetimes that are then used in e.g. the WMO/UNEP Ozone Assessments. It is not clear what 'assessed' means.

We have made this change (line 22).

2) Paragraph starting Line 60. This is the main paragraph where the ideas of uncertainty in lifetimes is introduced. It mentions different methods used to infer lifetimes, but does not clarify the 'time-varying' versus 'steady-state' which are used later with the same terminology (see text above). Moreover, it should be stated that it is the forward CCMs which give time time-varying values. There are a few other edits needed in this paragraph:

We have introduced 'time-varying' and 'steady-state' into this paragraph, discussed in line below.

Line 62. CCM is always chemistry-climate model (not chemical).

This has been corrected (line 55).

Line 66. WCRP is a proper name and officially written as Programme.

This has been corrected (line 59).

Line 66. 'Observations of surface concentrations are generally used only as a qualitative check in such calculations'. This is not true. For transient CCM simulations with surface mixing ratio boundary conditions these values are typically used as input and define the time dependence of the calculations. This was done for the SPARC CCMs for all gases including the three used here. I have looked at the SPARC report (and reference 9) and some flux boundary runs were done for CFC-11 and CFC-12, but those numbers did not feed into the recommended values. For steady-state calculations the surface vmr (or flux) would be irrelevant as the amount of tracer would cancel out in the burden/loss rate calculation.

This has been corrected (Lines 59-61):

CCMs use observed surface concentrations as input, and the accuracy of their modeled lifetimes largely depends on their ability to accurately model atmospheric transport and chemical loss processes.

Line 73. 'The second approach...'. It would be good to give this method a name because limitations of this approach are raised in the discussion. I believe that this is method (ii) in my paragraph above?

Yes, this is referring to the inverse modeling approach, we've added this to the text (line 68-69):

The second approach, referred to as the 'inverse modeling method', ...

Line 75. 'These different approaches have led to wide ranges in estimated lifetimes (see Table 1)': I don't think that this statement matches the previous text in this paragraph on the two methodologies and the idea of time-varying lifetimes. Aside from the new values from the submitted work (first two columns), Table 1 gives previous lifetime estimates from WMO (2003)/IPCC (2001), the SPARC (2013) steady-state estimate and Rigby et al (2013). I have not checked the source of WMO (2003) but I suspect that these are also steady-state values. I assume that Rigby et al (2013) is the Bayesian method? As noted above, the SPARC (2013) steady-state values are a combination of different methodologies.

We are trying to explain how the different methods, which have different assumptions have led to a wide range of lifetimes. This can be seen in our summary of some previous high profile recommended values in Table 1, along with the various lifetime estimates shown in Table 6.1 in SPARC. We've been more precise in this sentence (Line 71-79):

These different approaches include different assumptions and over the years have led to wide ranges in estimated steady-state lifetimes (see Table 1 of present manuscript and Table 6.1 of SPARC⁷), and therefore large differences in inferred emissions.

As for Rigby et al. (2013), we had stated in the previous sentence that they take Bayesian method with fixed emissions (Line 68 – 71):

The second approach, referred to as the 'inverse modeling method', infers lifetimes in a Bayesian framework using near-surface mole fraction measurements and, typically, fixed estimates of emissions as inputs to an atmospheric model¹⁰.

Line 76: 'therefore large differences in inferred emissions (Chapter 6 of SPARC⁷)'. I have looked and this Chapter does not give inferred emission. It only presents lifetimes. Need a different reference.

We changed the location of the reference to indicate that large uncertainties in lifetimes are reported in SPARC (line 78).

3) Line 92. 'different plausible lifetime'. Figure 1 focuses on the past with large growth in ODSs. The time-varying lifetime will be better suited to this than the steady-state lifetime. I would say that for

lifetimes the message from this figure is to use the time-varying ones. If the two different concepts were clear and named then this point could be made.

We agree that a steady-state lifetime is not suitable for estimating emissions over this time period, and that is one of the reasons that we believe our paper is valuable. This figure serves to underscore this point and hence caution future assessments from repeating this mistake. We have added some text to clarify this issue (Line 94 – 98):

With respect to lifetimes, we compare an assumed time-varying lifetime based on mean SPARC modeled values with an assumed steady-state lifetime of 52 years, as cited in WMO (2018), Table A-1³, to infer historical emissions. Hence, Figure 1 underscores the importance of using appropriate time-varying lifetimes in emissions and bank calculations.

4) Line 157 'SPARC CCM modelled...' Please add 'time-varying'.

We made this change.

5) Line 162 'set of models'. Change to 'set of CCMs' for clarity.

We made this change.

6) Line 162. 'lower than average CFC-11 lifetimes'. Might want to write this as singular: 'a lower than average CFC-11 lifetime'.

We did not make this change since it's not clear to us that writing this as singular is more accurate. We are talking about time-varying lifetimes here, and analysis shows that if a model is below average, it will have multiple years that are below average and hence, multiple lifetimes.

7) Line 178-179. Add words: 'using the SPARC CCM time-varying values'.

We made this correction.

8) Line 207 'constant scenario equal to the SPARC multi-model mean'. This is not clear. Do you mean a time-varying multimodel mean (e.g. as shown in Figure 4) or a single fixed value which is the time average of the multimodel mean? The first part of the sentence makes it sound like this constant scenario is not time varying.

We clarified this to indicate that the constant scenario is equal to the mean of the time-varying multi-model mean (Line 208-209):

which adopted both a time-varying lifetime equal to the SPARC multi-model mean, and a fixed lifetime scenario equal to the time-average mean of SPARC multi-model mean.

9) Line 218/219. 'are within the uncertainty of posterior observationally-derived emissions'. Could add 'our' before 'posterior' to make it clear that you mean your observationally derived emissions. Similarly could add 'our' before 'posterior' in line 221.

We have made this change.

10) Line 236/237 'shorter than the SPARC multi-model mean values'. Add 'time-varying'. Could also add 'throughout the time period'.

We have made this change.

11) Line 248. Change 'model' to 'CCM'.

We have made this change.

12) Line 250 'model'. Please be more specific with which model. 'Bayesian simulation model'? Also (line 251) I don't think DE is defined in the main text so write in full?

We have made these changes and added some clarifying text:

Line 247 – 248:

BPE model (i.e. not including the dependencies across molecules in the lifetime priors)

13) Line 266. 'The posterior time-averaged lifetime estimates and 2010 lifetime estimates...'. A bit confusing. These are both posterior. Change to e.g. 'The posterior time-averaged and 2010 lifetime estimates....'.

We have made this change.

14) Line 266. This is the first mention of the 2010 lifetime specifically in the main text. Please explain why you are showing the 2010 values (readers may not have looked at the Methods).

We have added a brief explanation here (line 262 – 265):

Our BPE analysis provides an estimate of time-varying lifetimes and emissions up until 2010, which is the end of the SPARC CCM simulations. For emissions beyond 2010, we adopt a constant lifetime using posterior emissions values from 2010 (see Methods).

15) Line 268. 'agrees with SPARC recommended...'. Need to say that this (in Table 1) is the SPARC steady-state lifetime (i.e. the only SPARC recommendation). At this point I think it is important for the readers to know the SPARC recommendation is not simply the SPARC CCM value, but is also based on other methods. I think you should also say here that your 2010 lifetime is the one you would expect to be comparable with the SPARC recommended value based on the methodologies.

We have updated this explanation by including a brief description of how the SPARC recommended lifetimes are computed (267 – 273):

To derive their recommended lifetimes, SPARC takes a weighted average of various estimation methods where the weights reflect the level of uncertainty in each method (see

SPARC Chapter 6 for details). Inverse-modeling and CCM derived lifetimes are among the methods used for all of CFC-11, 12 and 113. For CFC-11 and 12, satellite observation-derived lifetimes are also included. And for CFC-12 and 113, the tracer-tracer method is also included.

16) Line 273. 'The SPARC recommended lifetimes do not account for the intermolecule correlations...'. This is not completely true, but the picture is messy. The steady state CCM values that feed into the recommended values certainly do contain these correlations. The tracer-tracer correlation method (not discussed here) should have this information. The other methods, notably the inversion studies do not.

The relative variation of the new lifetimes derived here match the relative variation of the SPARC CCM values very well. The other methodologies, notably the inverse modelling give the different relative values that feed into the recommended numbers. The situation is more complicated than suggested. This paper is providing improved estimates from inverse modelling but these are now not independent of the CCM values. An improvement to the recommended values would still have to account for the other independent methodologies. It would be better to compare the new numbers with the inverse modelling first rather than discussing solely the SPARC recommended values.

As the reviewer says, the SPARC recommended values are messy and can only be briefly summarized here due to length constraints but can be found in the 250+ page SPARC report that is referenced. We have summarized the key points of the SPARC report to recognize that the CCM lifetime estimates are a component of the SPARC recommended values and explain this to the reader (shown below). However, we still do not believe the inter molecule correlations exhibited across CCMs are well captured since the SPARC recommended values are an average of the multi-model mean, with limited expert judgement on correlations between molecules (which does not yield absolute values but rather ratios of lifetimes). In our analysis we include both the means from the CCMs (which are part of the SPARC recommended values) but we also include the covariance structure between molecules calculated from all CCM models, which we do not believe is included in the SPARC recommended values. While it is true that the tracer-tracer method could also exhibit some of this inter molecule correlation, it is not included in the CFC-11 recommended value, so would only be relevant for CFC-12 and 113 dependencies, and still is only a fraction of the total average. We have added some text to explain the SPARC approach and limitations in this paragraph and refer the reader to Chapter 6 of SPARC for more details (Lines 265-296):

The posterior time-averaged lifetime and 2010 lifetime estimates are shown in Table 1 along with comparisons with previously published estimates, importantly the SPARC recommended steady-state lifetime. To derive their recommended lifetimes, SPARC takes a weighted average of various estimation methods where the weights reflect the level of uncertainty in each method (see SPARC Chapter 6 for details). Inverse-modeling and CCM-derived lifetimes are among the methods used for all of CFC-11, 12 and 113. For CFC-11 and 12, satellite observation-derived lifetimes are included, and for CFC-12 and 113, the tracer-tracer method are included. While our CFC-11 2010 lifetime estimates agree with the SPARC recommended lifetime estimates, our CFC-12 and 113 estimates are outside the 2-sigma range that SPARC estimated to be "most likely" (shown in Table 1), but within their "possible" 2-sigma range ((78, 151) for CFC-12 and (69, 138) for CFC-113; see Chapter 6 in SPARC). Why would our CFC-11 lifetime estimate agree with SPARC recommended

lifetimes, but not CFC-12 and 113? We attribute this to two factors. The first is that the SPARC recommended values do not explicitly account for the inter molecule correlations exhibited by the CCM modeled lifetimes. While the CCM derived lifetimes are a component of the weighted average, it is only the multi-model mean that is included and the inter-molecule correlations exhibited by the CCMs are not explicitly modeled in the weighted average. The second (which is a key limitation in earlier studies that use inverse-modeling to derive lifetimes, including SPARC) is that the recommended lifetimes depend on emissions assumptions through the inverse-modeling component of the estimate, whereas our analysis jointly infers emissions along with lifetimes. We expect that the SPARC prescribed emissions for CFC-12 and 113 are biased low relative to CFC-11 for the inverse-modeling component of their estimate, contributing to this difference in recommended values. This is supported by the relatively long lifetimes for CFC-12 and 113 coming from the inverse modeling component of the weighted average (see Table 6.1 from Chapter 6 of SPARC)

17) Line 275. 'key limitation to in'. Again, this is too general to be completely true. The forward CCM methodology does not depend on the emissions. The inverse modelling does depend on these. So, to be accurate, I think that the statement should be that key limitation applies to the inverse modelling method that feeds into the SPARC recommendation.

We have specified that we are referring to inverse-modeling approaches (Line 288 – 289):

which is a key limitation in earlier studies that use inverse-modeling to derive lifetimes,

18) Line 276. 'the recommend lifetimes depend on emission assumptions'. The recommended lifetimes are based on a combination of sources. As the recommended ones are 'steady state' the emissions are not relevant for the forward running CCMs; the steady-state lifetimes could be inferred with any emission value. For the observationally derived values in the SPARC report whether emissions are relevant will likely depend on the methodology.

The text added for the previous point clarifies that we are referring to inverse-modeling approaches.

19) Line 277-278 'We expect that the SPARC prescribed emissions for CFC-12 and 113 are biased low relative to CFC-11'. Following on from the comments above, I believe that this is only relevant for the inverse modelling contribution to the SPARC recommended lifetimes.

The paragraph has been substantially revised, and we have added additional text to clarify (291 – 296):

We expect that the SPARC prescribed emissions for CFC-12 and 113 are biased low relative to CFC-11 for the inverse-modeling component of their estimate, contributing to this difference in recommended values. This is supported by the relatively long lifetimes for CFC-12 and 113 coming from the inverse-modeling component of the weighted average (see Table 6.1 from Chapter 6, SPARC)

20) Line 301. Table 1. Please standardise the headings for columns 1 and 2. Both should say 'posterior'. State the time span for the average in column 2. Some information could be moved up to a heading which spans both columns so it is clear what the difference is (2010 versus time average).

This has been changed.

21) Line 459-460. 'lower than previous surface mole fraction trend-based estimates¹⁰'. Is that really the key point that these lifetimes are based on surface model fractions? They are based on inversions with a fairly simple global model. I think you need to relate this to the 'inverse modelling' methodology and use a consistent name.

When surface mole fraction data are used together with prescribed emissions estimates that may well be incorrect, then the inferred lifetimes are affected in previous inverse modelling studies. Our method uses much more information and allows for additional uncertainties (including in emissions). We believe the text is now clear regarding this.

To clarify the point raised by the reviewer, we have changed the language to (354 – 355):

This is an important distinction between this work and previous inverse-modeling studies

22) Line 464 'previous publications'. Specify previous inverse modelling or Bayesian estimation publications. (CCM papers and tracer-tracer correlations would contain this information).

We've changed the wording to say: previous inverse-modeling studies.

22) Line 509. 'lifetime of 57 years'. I know that it is not your paper (though one co-author overlaps), but having read the detail of lifetimes and got to this point, the reader will wonder why 57 years was used. Why not the SPARC/WMO recommended value? As the Montzka et al study is very high profile, and motivates this work, it would be helpful to say why they used this alternative value. This paragraph also compares the new emissions with those from Rigby et al (2019), but you don't say what lifetime was used there. Was that also 57 years? Please clarify that.

The reviewer has underscored a central motivation for this paper: the importance of lifetime assumptions in emissions and bank inference. Montzka et al. (2018) base their lifetime assumptions on two different CCM simulations, to be consistent with the 3D model that was used. Rigby et al. (2019) use SPARC recommended values in estimating global emissions. We've updated this sentence to include this (363 – 364) :

Montzka et al. (2018)⁴ used a single set of NOAA observations along with a constant lifetime of 57.5 years, derived from the two 3-D CCM simulations used in their analysis, to estimate

and Line 373 – 375:

... Note that Rigby et al. (2019) use the SPARC recommended lifetime values (52 years), thus their emissions estimates represent a lifetime-corrected version of Montzka et al. (2018) values.

23) Line 689 'SPARC modelled values'. Clarify this is from the SPARC CCMs – e.g. 'from the time varying SPARC CCM values'.

This wording has been adopted.

24) Line 690 'Compute the inverse of each modelled lifetime...'. Not really clear that you mean the inverse of each lifetime estimate for each molecule from the separate participating SPARC CCMs. Bullet point 2 gives the number of models but this could be stated higher up in this subsection.

We have changed this to (Line 513):

For each of the SPARC CCMs, compute the inverse of each modeled lifetime

25) Line 718 Add words 'SPARC time-varying CCM lifetime estimates....'

This has been changed

Reviewers' Comments:

Reviewer #3:

Remarks to the Author:

Thank you to the authors for incorporating my suggestions. I think that the paper is now much clearer (and more correct) in its discussion of lifetimes and the SPARC numbers.

The paper is now acceptable for publication, in my opinion.

While reading the revised pdf I did notice some small editorial things which I include here for information. (The line numbers are from the clean PDF).

Line 63. Maybe insert 'ODS' before 'lifetimes' to make it clear you are referring to this type of species.

Line 259-260. 'And for CFC-12 and 113...included'. This sentence does not read well (in my opinion). Could say: 'The tracer-tracer method provides values for CFC-12 and 113 based on an assumed CFC-11 lifetime'.

Line 278. Missing full stop.

Line 347. The response document said that you would add in 52 years for CFC-11 here, but it did not make it into the main paper. I think it is important for the discussion. Maybe could add '(e.g. 52 years for CFC-11)' as that is the main number but there are other species in Rigby et al.

REVIEWERS' COMMENTS

Reviewer #3 (Remarks to the Author):

Thank you to the authors for incorporating my suggestions. I think that the paper is now much clearer (and more correct) in its discussion of lifetimes and the SPARC numbers.

The paper is now acceptable for publication, in my opinion.

While reading the revised pdf I did notice some small editorial things which I include here for information. (The line numbers are from the clean PDF).

Thank you, we made these corrections.

Line 63. Maybe insert 'ODS' before 'lifetimes' to make it clear you are referring to this type of species.

We made this correction.

Line 259-260. 'And for CFC-12 and 113...included'. This sentence does not read well (in my opinion). Could say: 'The tracer-tracer method provides values for CFC-12 and 113 based on an assumed CFC-11 lifetime'.

We made this correction.

Line 278. Missing full stop.

We made this correction.

Line 347. The response document said that you would add in 52 years for CFC-11 here, but it did not make it into the main paper. I think it is important for the discussion. Maybe could add '(e.g. 52 years for CFC-11)' as that is the main number but there are other species in Rigby et al.

We made this correction.